# Vision Foundation Model Enables Generalizable Object Pose Estimation

**Kai Chen[1], Yiyao Ma[1], Xingyu Lin[2], Stephen James[3], Jianshu Zhou[1]**
**Yun-Hui Liu[1], Pieter Abbeel[2], Qi Dou[1]**
[1]The Chinese University of Hong Kong, [2]UC Berkeley, [3]Dyson Robot Learning Lab

## Abstract

Object pose estimation plays a crucial role in robotic manipulation, however, its practical applicability still suffers from limited generalizability. This paper addresses the challenge of generalizable object pose estimation, particularly focusing on category-level object pose estimation for unseen object categories. Current methods either require impractical instance-level training or are confined to predefined categories, limiting their applicability. We propose VFM-6D, a novel framework that explores harnessing existing vision and language models, to elaborate object pose estimation into two stages: category-level object viewpoint estimation and object coordinate map estimation. Based on the two-stage framework, we introduce a 2D-to-3D feature lifting module and a shape-matching module, both of which leverage pre-trained vision foundation models to improve object representation and matching accuracy. VFM-6D is trained on cost-effective synthetic data and exhibits superior generalization capabilities. It can be applied to both instance-level unseen object pose estimation and category-level object pose estimation for novel categories. Evaluations on benchmark datasets demonstrate the effectiveness and versatility of VFM-6D in various real-world scenarios. Project website: vfm-6d.github.io/.

## 1 Introduction

Object pose estimation refers to the procedure of identifying the position and orientation of target objects. This essential operation is highly valued in various scenarios such as robotic manipulation [1, 2], augmented reality [3, 4], and embodied intelligence [5, 6], etc. Predominantly, most existing approaches [7, 8] largely rely on instance-level training to associate object frame and camera frame for object pose estimation. These methods struggle to accurately predict the poses of objects that were not present during the training phase. In order to enhance the manipulation and interaction with a broad range of objects, there is a considerable demand for generalizable object pose estimation.

Towards generalizable object pose estimation, two main streams of effort have been made in recent literature: instance-level unseen object pose estimation [9, 10, 11, 12] and category-level object pose estimation [13, 14, 15]. Instance-level unseen object pose estimation methods work by comparing the observed object to instance-level reference images and then estimating a relative pose from the reference to the observation. These methods typically necessitate either textured CAD models or the collection and labeling of reference images for each object instance. However, these requirements are often impractical for many real-world scenarios [16, 17, 18], such as open-world mobile manipulation, where the objects that the agent encounters are unpredictable. This greatly hampers the practical applicability of instance-level unseen object pose estimation methods. On the other hand, recent category-level approaches learn to recover object pose from category-level object semantic and shape features. By learning category-level characteristics from data, these methods can be readily applied to novel objects that belong to a certain category, provided the model is trained on that category.

38th Conference on Neural Information Processing Systems (NeurIPS 2024).

However, these category-level approaches fall short when encountering objects from an unseen category. For example, a model trained solely on 'mug' would be unable to estimate the pose for a 'laptop'. The generalization capability of these methods is thus limited by a set of pre-defined object categories, which is far from meeting the demands of practical applications.

These existing approaches inspire us to consider the following question: *Can we perform category-level object pose estimation for an unseen object category?* The goal is to leverage the complementary advantages of both types of methods to achieve generalizable object pose estimation. Category-level estimation significantly reduces inference costs for unseen object instances by eliminating the need for extensive scanning of textured CAD models and labor-intensive pose annotations for multi-view reference images. Furthermore, if the method is not limited to predefined categories, it would exhibit high generalization capability for objects in any category. Achieving this challenging goal requires an object representation that is robust to object intra-class variation and also generalizable to novel object categories. Recent advancements in vision and language models [19, 20, 21, 22] have shown that their robust object representation can effectively enhance the generalization capability of various downstream tasks [23, 24, 25, 26]. However, how could these models be applied to generalizable object pose estimation? It is still an open question that remains underexplored.

We explore this question by innovating a vision foundation model based framework VFM-6D for generalizable object pose estimation. Given a query image of the target object, VFM-6D elaborates the scheme of object pose estimation into two stages: foundation model based object viewpoint estimation and foundation model based object coordinate map estimation. The object pose and size then are jointly optimized based on the dense correspondences derived from the predicted object coordinate map. To accommodate to novel object categories, VFM-6D resorts to category-level reference images for object viewpoint estimation and coordinate map estimation. The viewpoint of the query object is identified by matching the query image with multi-view references, and the query coordinate map is estimated by matching the shape of the query object with that of the reference. However, precisely matching the query and its category-level references can be challenging due to potential differences in textures and shapes, even when utilizing a vision foundation model. Taking the pre-trained DINO-v1 model [20] as an example, as shown in Fig. 1, the pre-trained model struggles to match the query image with its closest reference image. The cosine similarity derived from pre-trained DINO-v1 often appears to be relatively indistinguishable.

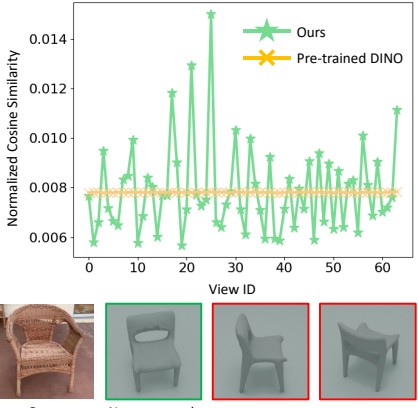

Figure 1: Results with pre-trained DINO-v1 [20]. We can observe that directly using the pre-trained DINO-v1 cannot identify the nearest viewpoint from the feature cosine similarity. Our proposed feature lifting module can significantly improve the differentiability of multi-view object representations and identify the most similar viewpoint precisely.

In this paper, we propose to address the above issue by innovating a 2D-to-3D feature lifting module, which leverages the 3D position embedding of objects to effectively adapt the vision foundation model to extract view-aware object representations for precise query-reference matching and object viewpoint estimation. In addition, to ensure reliable coordinate map estimation for a diverse range of query objects, we further propose an effective shape-matching module. This module enhances the object shape representation by harnessing robust semantics from the pre-trained vision foundation model. As a result, we can effectively match the query object with the reference one, even in the face of potential intra-class shape variations. To effectively adapt the pre-trained vision foundation model to the task of object pose estimation, we freeze the foundation model backbone and fine-tune the proposed feature lifting and shape-matching modules using cost-effective synthetic data. Thanks to the elaborated two-stage framework, the robust object representation from the pre-trained vision foundation model, and the improvement introduced by our proposed feature lifting and shape-matching modules, VFM-6D exhibits superior generalization capabilities. As depicted in Fig. 2, VFM-6D can be broadly applied to both instance-level unseen object pose estimation and category-level object pose estimation for novel categories, using either scanned object shape templates or object shapes generated via text-to-3D generation. We summarize our contributions as follows:

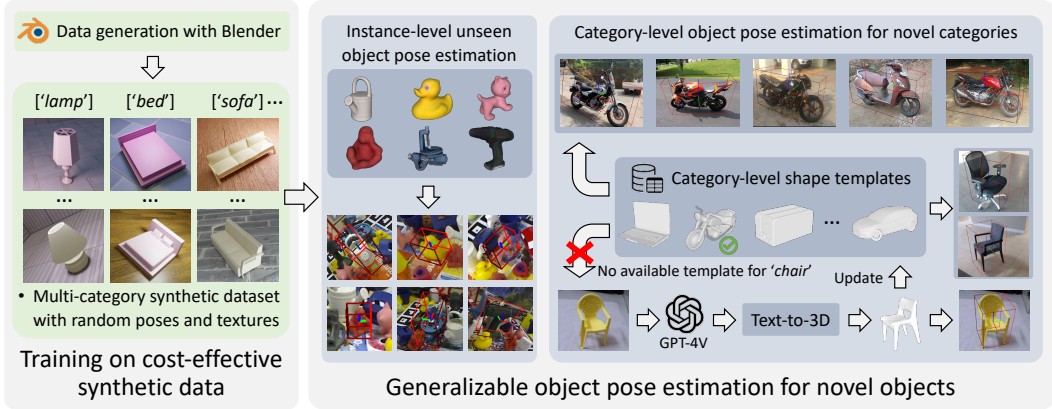

Figure 2: Our proposed VFM-6D is highly generalizable. After training on cost-effective synthetic data, it can be widely applied to instance-level unseen object pose estimation and category-level object pose estimation for novel categories.

- We propose VFM-6D, a novel framework that elaborates object pose estimation into foundation model based object viewpoint estimation and object coordinate map estimation, for effective and generalizable object pose estimation.

- We propose a novel 2D-to-3D feature lifting module to improve the capability of existing vision foundation models to extract view-aware object representations for precise query-reference matching and object viewpoint estimation.

- We present a robust shape matching module, which enhances the object shape representation by harnessing robust semantics from the pre-trained vision foundation model and enables reliable coordinate map estimation for a diverse range of objects in different shapes.

- We demonstrate that by training VFM-6D on cost-effective synthetic data, we can effectively apply 2D vision foundation models to the task of 6D object pose estimation. Extensive evaluations on representative benchmark datasets Wild6D, CO3D and LINEMOD demonstrate the superior generalization capability of VFM-6D.

## 2 Related Works

**Unseen Object Pose Estimation.** The current methods primarily employ a matching-based paradigm. OnePose [27, 28] utilizes a video scan to reconstruct a SfM model and match the query image to estimate the object pose. Gen6D [9] bypasses the need for SfM by using pre-posed reference images to match the query image and refine the pose. SA6D [10] builds on Gen6D by enhancing pose accuracy and robustness, particularly in occlusion scenarios, using RGB-D reference images. RelPose [29, 30] samples multiple pose hypotheses and verifies the correct pose based on an energy function conditioned on the similarity between reference and query images. Moreover, MegaPose [11] employs the CAD model to generate reference images and estimates the pose by comparing the rendered and observed images. FoundationPose [31] eliminates the need for CAD by using instance-level reference images from multiple viewpoints. SAM-6D [32] establishes 3D-3D correspondences to calculate the pose of each valid proposal identified by Segment-Anything-Model. FoundPose [33] further resorts to bag-of-words descriptors for unseen object pose estimation. However, all these methods require instance-level reference images or CAD models. Our method differs from them by using category-level reference images, which can be reused for objects belonging to the same category and significantly reduce the burden of collecting references for each new object instance.

**Category-Level Object Pose Estimation.** The main challenge in category-level object pose estimation is the significant intra-class variation observed among different objects. Existing methods for addressing this challenge include canonical object representation [15, 34], robust pose formulation [35, 36], and category-relevant feature extraction [37, 38]. More specifically, Wang et al. [13] propose to estimate pose parameters within a normalized object space. Tian et al. [39] leverages a template point cloud to explicitly reconstruct the 3D shape for the novel object, and then register the object point cloud with the reconstructed object shape for category-level object pose estimation. To

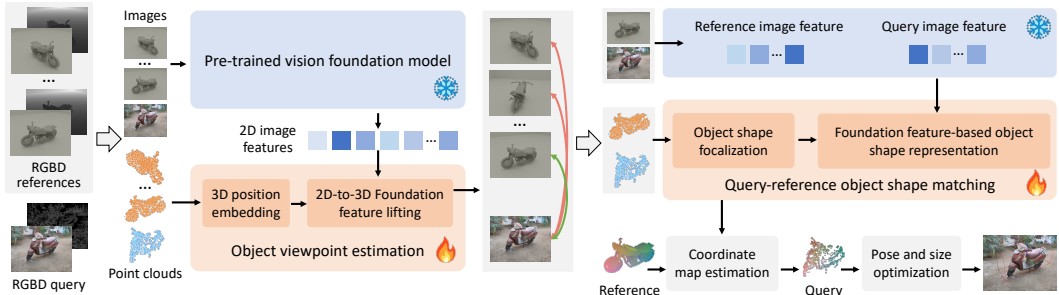

Figure 3: An overview of the proposed VFM-6D framework for generalizable object pose estimation.

improve pose accuracy for instances with shape variations, Chen et al. [40] dynamically adjust the template feature with instance semantics, and Lin et al. [41] use an adaptive keypoint-based method to explicitly consider local-global geometry information. Recently, Zhang et al. [42] applied a score-based diffusion model for accurate category-level object pose estimation and tracking. However, most of these methods rely on category-specific training, and cannot be applied to novel object categories.

**Object Pose Estimation in Robotics.** As an important scheme to perceive an object's status, object pose estimation has wide applications in various robot scenarios. For example, Wen et al. [43] use the object pose to transform object-centric grasping configuration from object frame to robot frame for robotic grasping. Geng et al.[44] utilize pose information from object parts to enable generalizable object manipulation. Morgan et al.[45] employ high-precision object pose tracking for closed-loop vision-based robot control. An et al. [46] leverage object pose feedback to plan the robot's motion trajectory, ensuring accurate and efficient robot manipulation. Li et al. [47] apply object-centric pose prediction to stimulate the reasoning ability of Multimodal Large Language Models in robot manipulation tasks. Despite the usefulness of object pose for robot perception, its practical application is limited by its restricted generalization capability. In this paper, our aim is to propose a novel method that enhances the perception of object pose in a more flexible and generalizable manner.

**Vision Foundation Model.** By pre-training at scale, the recent vision foundation model manages to produce quite robust and expressive representations for diverse objects, significantly boosting the generalization performance of various vision tasks, including recognition [23, 48], detection [49, 24], and segmentation [50, 25, 26]. In the context of object pose estimation, Zhang et al. [51] use a pre-trained DINO to extract consistency constraints from images in the wild for self-supervised category-level object pose estimation. Goodwin et al.[52] empirically select features from DINO layers to match sparse keypoints for zero-shot category-level object pose estimation. Chen et al. [53] directly fuse the object-specific hierarchical geometric features with semantic DINO-v2 features to enhance the robustness of category-level pose estimation. OV9D [54] uses DINO-v2 and a text-to-image diffusion model to infer the object coordinate map and estimate the pose of a target instance via pose fitting. In this work, we also hope to leverage the robust representation from the foundation model. Moreover, we explore a cost-effective way to further adapt the pre-trained feature with task-specific data and we demonstrate that it can significantly improve the performance for generalizable object pose estimation.

## 3 Methodolody

### 3.1 Overview of VFM-6D

We present VFM-6D, which aims to tackle the problem of generalizable object pose estimation via category-level object pose estimation for novel object categories. To achieve this challenging target, VFM-6D is built upon DINO-v2 [21] and elaborates the object pose estimation scheme into two stages: object viewpoint estimation and object coordinate map estimation. VFM-6D jointly leverages object RGB image and partial point cloud derived from depth map for object pose estimation. In the first stage, we determine the initial object 3D rotation with respect to the camera frame. In the second stage, we estimate a coordinate map for the object, which establishes dense correspondences between the object and camera frames for object pose estimation. Inspired by [9, 12, 55], both stages exploit a reference-based design. Let $\mathcal{O}_q = (\mathcal{I}_q, \mathcal{X}_q)$ be the query object observation[1], and

---

[1] $\mathcal{I}$ and $\mathcal{X}$ respectively represent the object image and the object point cloud.

$\{\mathcal{O}_r^i = (\mathcal{I}_r^i, \mathcal{X}_r^i)\}_{i=1}^k$ be its corresponding multiview references with known poses $\{\mathbf{R}_r^i | \mathbf{t}_r^i\}_{i=1}^k$. As shown in Fig. 3, VFM-6D identifies the object viewpoint by image-level matching between $\mathcal{O}_q$ and $\{\mathcal{O}_r^i\}_{i=1}^k$. Then, VFM-6D further leverages point-level shape matching between the query and reference objects for coordinate map estimation. Without loss of generality, we exploit the coordinate map defined in the normalized object coordinate space (NOCS [13]) in this work.

Recall that VFM-6D is designed for category-level object pose estimation of novel object categories. It aims to address the challenge of intra-class variation in query-reference matching by utilizing the robust object representations derived from the vision foundation model. In this regard, we first introduce a novel foundation feature lifting module in Sec. 3.2 for image matching and viewpoint estimation. Based on the estimated object orientation and the matched reference image, we subsequently present a foundation feature-based object shape representation in Sec. 3.3, which resorts to object semantics from foundation models for robust shape matching and NOCS coordinate map estimation across diverse object categories. Finally, in Sec. 3.4, we present how to fine-tune the VFM-6D framework with cost-effective synthetic data for generalizable object pose estimation.

## 3.2 Query-reference Image Matching for Object Viewpoint Estimation

VFM-6D estimates the viewpoint of $\mathcal{O}_q$ by identifying the closest reference image from $\{\mathcal{O}_r^i\}_{i=1}^k$. Following [56], the image distance is measured based on the cosine similarity of object features extracted from the corresponding images. In order to handle a variety of novel object categories, VFM-6D leverages the robust object representation provided by the vision foundation model. However, this model falls short when it comes to estimating the viewpoint of an object. As depicted in Fig. 1, the query image often exhibits a very similar cosine similarity with its multiview reference images, making it challenging to effectively identify the closest reference for $\mathcal{O}_q$. This scenario underscores the limited 3D representation capabilities of the visual foundation model pre-trained with 2D image and text data. Such models tend to extract features that are less sensitive

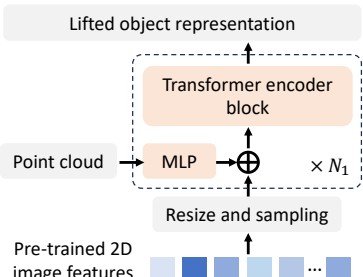

Figure 4: 2D-to-3D foundation feature lifting for view-aware object representation.

to changes in viewpoint, for the sake of semantic robustness. To address this issue, we propose an effective foundation feature lifting module for object viewpoint estimation. As depicted in Fig. 4, the feature lifting module is designed to enhance the discriminative capacity of the foundation model's object representation. It is accomplished by lifting the object representation from 2D to 3D, utilizing the 3D positional information encapsulated within object point clouds. Specifically, let $\mathcal{F}$ represent the pre-trained object representation from the vision foundation model. The feature lifting module begins by encoding the 3D object position from $\mathcal{X}$ using a two-layer MLP, thereby encapsulating valuable 3D position information along with contextual details. Following this, it lifts $\mathcal{F}$ by integrating the extracted 3D position embedding into $\mathcal{F}$ via a Transformer encoder block as $\mathcal{F}' = \texttt{TransformerEncoder}(\mathcal{F} + \texttt{MLP}(\mathcal{X}))$. Based on the lifted object feature, we match the query image with the reference image that has the highest cosine similarity.

## 3.3 Query-reference Shape Matching for Coordinate Map Estimation

Let $\mathcal{O}_r$ be the matched reference for $\mathcal{O}_q$, and $\{\mathbf{R}_r | \mathbf{t}_r\}$ be the reference object pose. In this section, we perform point-level shape matching between $\mathcal{O}_q$ and $\mathcal{O}_r$ to estimate the NOCS coordinate map for $\mathcal{O}_q$. Overall, the coordinate map estimation is carried out in two steps: foundation feature based object shape representation and shape matching for coordinate map estimation. We denote the query and reference object shape as $\mathcal{X}_q \in \mathbb{R}^{n \times 3}$ and $\mathcal{X}_r \in \mathbb{R}^{m \times 3}$, which are two partial object point clouds derived from the corresponding depth map. $\mathcal{X}_q$ and $\mathcal{X}_r$ would vary dramatically in the camera space. In order to effectively model object shape from them, inspired by [57], we propose to first use the object viewpoint $\mathbf{R}_r$ that we have estimated in Sec. 3.2 to focalize $\mathcal{X}_q$ and $\mathcal{X}_r$ into a canonical and normalized coordinate space. Specifically, the focalization step could be formulated as $\mathcal{X}'_{(q,r)} = \mathbf{R}_r^{-1} \times \frac{\mathcal{X}_{(q,r)} - \bar{\mathcal{X}}_{(q,r)}}{\max\|\mathcal{X}_{(q,r)} - \bar{\mathcal{X}}_{(q,r)}\|}$, where $\bar{\mathcal{X}}$ denotes the center of the original partial point cloud in the camera frame. Based on the focalized object point cloud, we then integrate the point cloud based shape encoding with the robust image semantics of the vision foundation model for object shape representation. As shown in Fig. 5, we use a point cloud Transformer (PCT) [58] for object shape encoding.

The PCT will first patchify the forcalized point clouds with $K$ nearest neighbor points and then exploit a Transformer-based encoder-decoder to extract geometry features for point cloud based shape encoding. Then, we further integrate the pre-trained image feature with the point cloud encoding via a Transformer encoder block. Usually, the pre-trained image feature from the vision foundation model exhibits high robustness for objects in the same category. For example, object patches in the same semantics would have similar embeddings [20]. Our proposed shape representation module therefore leverages this robust semantic relationship to bridge the shape gap between query and reference and enhances their point cloud based representations for robust shape matching. Let $\mathcal{G}_q \in \mathbb{R}^{n \times d}$ and $\mathcal{G}_r \in \mathbb{R}^{m \times d}$ be the pointwise shape embedding of $\mathcal{O}_q$ and $\mathcal{O}_r$ respectively. The NOCS coordinates for the query are then estimated by $\mathcal{M}_q = \texttt{Softmax}(\mathcal{G}_q \times \mathcal{G}_r^\top) \times \mathcal{M}_r$, where $\mathcal{M}_r \in \mathbb{R}^{m \times 3}$. Note that $\mathcal{M}_r$ represents the NOCS coordinates for the reference object. It is generated based on the template object shape and ground-truth pose for each reference image, which is assumed to be available in VFM-6D.

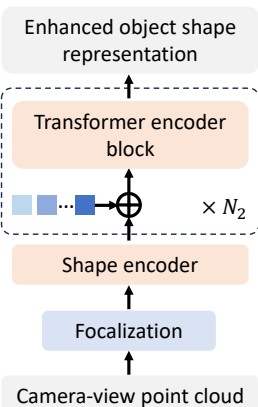

Figure 5: Foundation feature based object shape representation.

## 3.4 Training VFM-6D with Cost-effective Synthetic Data

In this section, we present training details of VFM-6D. Specifically, we freeze the vision foundation model backbone and only train the proposed 2D-to-3D foundation feature lifting module and the foundation feature based object shape representation module.

**Synthetic data generation.** We customized the synthetic data generation pipeline from [59] to utilize Blender for generating photorealistic synthetic data, which we then employed for cost-effective model training. To ensure diversity in the synthetic training data, we collected 20 categories of objects from ShapeNet [60]. Each category contains 6 unique object instances, resulting in a total of 120 distinct object assets. For each category, one object instance was randomly selected to synthesize a total of 10K RGBD images. Concurrently, the object mask and ground-truth object coordinate map were generated for each image. In total, the synthetic training dataset comprises 200K RGBD images. For more detailed information about the synthetic training data, please refer to the appendix.

**Training objective for viewpoint estimation.** For each object category, we randomly select one object instance as the shape template and take the remaining five instances as query objects during training. For object viewpoint estimation, we adopt the pose-aware contrastive loss in [56], which considers the negativeness of each reference image and fully harnesses the positive and negative query-reference pairs to train the foundation feature lifting module:

$$\mathcal{L}_{view} = -\log \frac{\exp(f_q \cdot f_{r+}/\tau)}{\sum_{i=1}^{k} d(\mathbf{R}_q, \mathbf{R}_r^i)\exp(f_q \cdot f_r^i/\tau)}, \tag{1}$$

where $f_{q,r} = \texttt{AvgPool}(\mathcal{F}'_{q,r})$ denotes the global object representation after feature lifting, $r+$ indicates the closest reference image to the query one, $\tau$ is a temperature parameter. Moreover, $d(\cdot)$ is defined as the normalized viewpoint difference between query and reference, which is computed as $d(\mathbf{R}_q, \mathbf{R}_r^i) = \arccos(\frac{\text{tr}(\mathbf{R}_q^\top \mathbf{R}_r^i)-1}{2})/\pi$. We also follow [61] to handle symmetrical objects.

**Training objective for coordinate map estimation.** For coordinate map estimation, we compare the predicted query coordinate map with the ground-truth one. For symmetrical objects, we adopt the Chamfer Distance loss to constrain the predicted coordinate map: $\mathcal{L}_{coor} = \text{CD}(\mathcal{M}_q, \mathcal{M}_{gt})$. For asymmetrical objects, we measure the coordinate map distance via a smooth L1 loss:

$$\mathcal{L}_{coor} = \sum_{p_1,p_2} \begin{cases} 0.5(p_1 - p_2)^2/\beta, & \text{if } |p_1 - p_2| < \beta, \\ |p_1 - p_2| - 0.5 \times \beta, & \text{otherwise}, \end{cases} \tag{2}$$

where $p_1 \in \mathcal{M}_q$ and $p_2 \in \mathcal{M}_{gt}$ respectively.

The total training loss is $\mathcal{L} = \mathcal{L}_{view} + \mathcal{L}_{coor}$. By training VFM-6D with the synthetic data, we effectively adapt the existing vision foundation model to generalizable object pose estimation without incurring heavy costs related to real-world data collection and pose annotation.

Table 1: Category-level object pose estimation results on 5 novel categories of Wild6D.

| Method | Unseen | $5°2cm$ | $5°5cm$ | $10°2cm$ | $10°5cm$ |
|---|---|---|---|---|---|
| SPD [39] | ✗ | 2.6 | 3.5 | 9.7 | 13.9 |
| SGPA [40] | ✗ | **20.1** | **27.8** | 29.0 | 39.4 |
| DualPoseNet [66] | ✗ | 17.8 | 22.8 | 26.3 | 36.5 |
| GPV-Pose [67] | ✗ | 14.1 | 21.5 | 23.8 | 41.1 |
| PoseContrast [56] | ✔ | 2.3 | 4.7 | 5.5 | 10.1 |
| ZSP [52] | ✔ | 9.6 | 12.1 | 16.6 | 23.0 |
| Ours | ✔ | 19.3 | 21.6 | **34.9** | **44.2** |

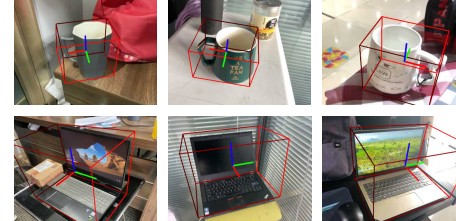

Figure 6: Qualitative results of VFM-6D on '*mug*' and '*laptop*' of Wild6D.

## 4 Experiments

### 4.1 Dataset and Setup

We mainly evaluated VFM-6D on two category-level benchmark datasets Wild6D [62] and CO3D [63], to investigate its generalization capability for category-level object pose estimation of novel object categories. Wild6D testing data consists of 162 different object instances in 5 typical table-top object categories (*i.e.*, '*laptop*', '*camera*', '*mug*', '*bottle*', and '*bowl*'). CO3D is more comprehensive, which covers thousands of object instances in 50 categories varying from vehicles (*e.g.*, '*motorcycle*' and '*bicycle*') to daily necessities (*e.g.*, '*backpack*' and '*handbag*'). Apart from category-level evaluation, we wonder whether VFM-6D is also applicable to the setting of instance-level unseen object pose estimation. We therefore further evaluated VFM-6D on LINEMOD [64], which is a representative instance-level dataset involving 13 textureless object instances.

For the experiment on CO3D, we followed [52] to evaluate VFM-6D on 20 categories with unified and precise object pose labels. It is worth noting that all object categories involved in evaluation are not used during the training stage. For the category-level benchmark evaluation, we collected the corresponding untextured category-level shape templates from the publicly available 3D object assets [60] ,or generated the shape template via text-to-3D [65] based on the category name. For the evaluation of instance-level unseen object pose estimation, we took each object instance as a category and leveraged its instance-level CAD model as the shape template. To synthesize the multiview reference images required by VFM-6D, we normalized the shape template to a unit diameter. We then uniformly sampled $k$ camera viewpoints from the sphere's surface centered on the template and rendered an RGB-D reference image at each viewpoint using Blender.

### 4.2 Evaluation Metrics

We followed the evaluation protocols that have been used on different benchmark datasets to evaluate the object pose estimation accuracy of VFM-6D. Specifically, we followed [62, 51] to report the mean Average Precision for different thresholds of $a°bcm$ to evaluate the performance on Wild6D. Considering that the size of objects in CO3D varies greatly, from several meters (*e.g.*, '*car*' and '*motorcycle*') to a few centimeters (*e.g.*, '*remote*' and '*mouse*'), we followed [52] to focus on the evaluation of object rotation and reported the accuracy for different rotation thresholds (*i.e.*, ACC.15° and ACC.30°). We followed [68, 69, 70] to exploit a model-based evaluation metric ADD for instance-level experiments on LINEMOD, and reported the recall of ADD that is smaller than 0.1 of the object diameter (*i.e.*, ADD-0.1d).

### 4.3 Main Results

**Wild6D.** Table 1 presents the evaluation results of category-level object pose estimation on Wild6D. Specifically, we first compared VFM-6D with four conventional category-level approaches. Since these conventional methods necessitate training on target categories, we trained their models on NOCS [2] dataset [13] and then assessed their performance on Wild6D. In addition, we compared VFM-6D with two category-agnostic approaches, PoseContrast [56] and ZSP [52]. As shown in Table 1, without training on the 5 target categories, our proposed VFM-6D significantly surpasses the competing two category-agnostic methods. Moreover, it achieves a comparable or even superior pose accuracy when compared with conventional methods, which however are heavily dependent on category-specific training. Fig. 6 showcases qualitative pose prediction results of VFM-6D across different object categories of Wild6D.

---

[2]NOCS is a mixture of synthetic and real-world dataset, covering the 5 object categories depicted in Wild6D.

Table 2: Category-level object pose estimation results on 20 unseen categories of CO3D dataset. We report Acc.15° / Acc.30° averaged across all 20 categories. We also report results for an illustrative subset of categories. Please refer to the appendix for full per-category results.

| Method | Motorcycle | Backpack | Bicycle | Teddybear | Car | Chair | Average |
|---|---|---|---|---|---|---|---|
| LoFTR [71] | 4.2 / 10.9 | 1.1 / 2.8 | 2.2 / 7.8 | 12.5 / 18.9 | 30.0 / 36.2 | 8.9 / 15.2 | 8.4 / 13.7 |
| LightGlue [72] | 4.2 / 11.8 | 0.5 / 4.9 | 14.4 / 27.0 | 8.2 / 19.5 | 7.5 / 13.8 | 3.5 / 8.2 | 5.9 / 12.7 |
| GeDi [73] | 4.3 / 11.5 | 0.5 / 2.7 | 5.6 / 17.6 | 9.4 / 25.7 | 0.5 / 3.0 | 10.5 / 23.0 | 5.3 / 13.4 |
| ZSP [52] | 25.0 / 49.2 | 4.4 / 13.0 | 27.0 / 49.3 | 23.7 / 43.4 | 61.8 / 69.3 | 30.0 / 41.6 | 27.1 / 44.7 |
| Ours | 56.4 / 76.3 | 30.6 / 47.4 | 46.5 / 59.2 | 25.2 / 54.4 | 55.6 / 74.4 | 72.1 / 86.8 | **50.2 / 67.4** |

Table 3: Instance-level object pose estimation results measured by ADD-0.1d on LINEMOD dataset. We report average score over all instances and per-instance score for an illustrative subset of instances. Please refer to the appendix for full per-instance results.

| Method | ape | bench. | camera | can | cat | driller | duck | Average |
|---|---|---|---|---|---|---|---|---|
| LatentFusion [68] | 88.0 | 92.4 | 74.4 | 88.8 | 94.5 | 91.7 | 68.1 | 87.1 |
| OSOP [69] | 86.1 | 94.6 | 69.3 | 80.4 | 88.0 | 79.5 | 92.7 | 82.0 |
| FS6D [70] | 78.0 | 88.5 | 91.0 | 89.5 | 97.5 | 92.0 | 75.5 | **91.5** |
| Ours | 88.5 | 91.9 | 97.2 | 74.9 | 96.4 | 85.1 | 88.0 | 90.3 |

**CO3D.** We conducted more comprehensive evaluation on CO3D. Table 2 presents comparative results against typical category-agnostic approaches, including two image feature point matching methods LoFTR [71] and LightGlue [72], one point cloud registration method GeDi [73], and the recent vision foundation model based method ZSP [52]. For all the competing methods, we exploited the same category-level reference images for object pose estimation. For LoFTR, LightGlue and ZSP, the object point cloud are further leveraged to lift the extracted 2D matches into 3D for object pose estimation. As shown in Table 2, both image feature point matching and point cloud registration methods fall short when being applied to match objects in different textures and shapes. We also observed that the matching results of ZSP are spatially biased, concentrating on some specific parts, *e.g.*, the wheels of the bicycle and the wings of the toyplane. Its incomplete and noisy matches limit the final pose accuracy. In contrast, our proposed VFM-6D achieves a higher accuracy and consistently outperforms all competing methods by a large margin. Please refer to the appendix for qualitative results of VFM-6D on CO3D.

**LINEMOD.** Table 3 presents the comparative results against three baseline approaches. LatentFusion [68] and OSOP [69] are two zero-shot methods for instance-level unseen object pose estimation. FS6D is a few-shot method, which requires finetuning on the LINEMOD dataset using a limited number of instance samples (*i.e.*, 16 shots). As can be observed, although our proposed VFM-6D is designed for category-level object pose estimation, it is also applicable to the instance-level setting, if we take each object instance as one object category. VFM-6D outperforms two zero-shot baselines and achieves comparable performance to the few-shot baseline. This experiment highlights the practicality of VFM-6D. It is capable of leveraging the precise instance-level CAD models for object pose estimation. Assuming that the agent encounters a novel object in the environment, VFM-6D is able to estimate the object pose based on its CAD model when it is available. Otherwise, VFM-6D can estimate the object pose based on its category information.

## 4.4 Ablation Studies

**Effects of individual modules.** To assess the efficacy of each module within VFM-6D, we conducted evaluations using different design alternatives. First, we substituted the proposed 2D-to-3D feature lifting module with foundation feature-based template matching [74] for object viewpoint estimation. Second, we eliminated the proposed object foundation model-based shape representation module and employed a similar method to [52] to directly estimate the query coordinate map based on pixel-wise foundation feature similarity. We replaced the original module of VFM-6D separately and evaluated the performance on CO3D. Table 4 displays the average accuracy over a subset of five categories from CO3D. For a more detailed view of the results, please refer to the appendix. As we can see, both modules contribute substantially to enhancing pose accuracy. Compared with conventional template matching, our proposed foundational feature lifting module demonstrates superior precision and robustness in object viewpoint estimation. The proposed shape representation module effectively aligns the query and reference object shapes, thereby refining the accuracy of the predicted query coordinate map used for object pose estimation.

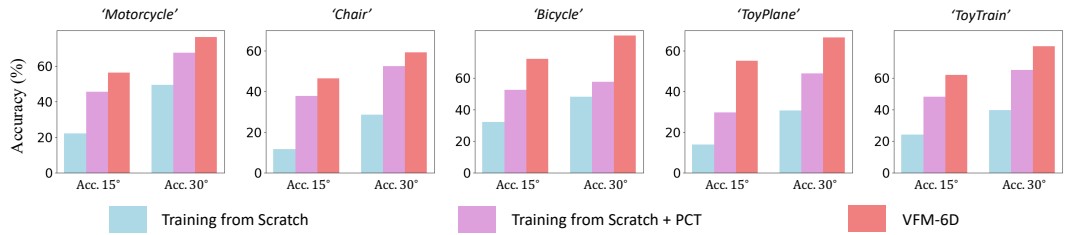

Figure 7: Comparison between VFM-6D and two training-from-scratch alternatives.

Table 4: Ablation study of individual modules of VFM-6D. The average accuracy over *'motorcycle'*, *'bicycle'*, *'chair'*, *'toyplane'*, and *'toytrain'* are reported.

| Settings | Acc.15° | Acc.30° |
|---|---|---|
| w/o feature lifting | 48.8 | 62.9 |
| w/o shape module | 48.6 | 66.7 |
| w/o both | 35.2 | 53.0 |
| Ours | **58.4** | **73.8** |

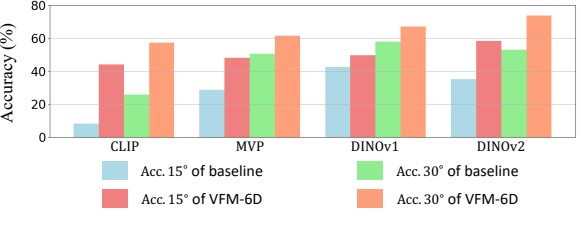

Figure 8: Results of VFM-6D with different vision foundation models.

**Effects of vision foundation model.** In this experiment, we aim to answer the following two questions: (1) Does the pre-trained vision foundation model improve the performance of generalizable object pose estimation? (2) How about the performance when applying different vision foundation models to VFM-6D? To answer the first question, we compared VFM-6D with two additional alternatives: (i) Training the whole VFM-6D from scratch. (ii) Replacing the adopted vision foundation model with a point cloud Transformer (PCT) [58], and training the whole framework from scratch. Note that the training of two alternatives is based on the same synthetic data as training VFM-6D. Fig. 7 presents the evaluation results. As can be observed, in training-from-scratch setting, the point cloud based alternative (*i.e.*, +PCT) exhibits a higher generalization capability. By pre-training at scale, the vision foundation model produces robust semantic features for various objects, which is important for generalizable object pose estimation.

To answer the second question, we evaluated VFM-6D with additional vision foundation models, including CLIP [19], MVP [75], and DINO-v1 [20]. For each vision foundation model, we also compared its performance with the corresponding baseline without our proposed feature lifting and shape matching modules. Fig. 8 depicts the evaluation results. As can be observed, our proposed VFM-6D can generally boost the object pose estimation performance, utilizing object representations from different vision foundation models. Thanks to the advanced pre-training strategies adopted in DINO-v2, this model produces high-quality visual feature maps, which also help to improve the pose accuracy and achieve the best performance.

**Effects of number of reference images.** We conducted experiments with different numbers of reference images varying from 16 to 96, and compared the object pose accuracy before and after the query-reference object shape matching and coordinate map based object pose estimation. Fig. 9 depicts the evaluation results on a subset of five categories from CO3D (*i.e.*, *'motorcycle'*, *'bicycle'*, *'chair'*, *'toyplane'*, and *'toytrain'*). In Fig. 9, 'baseline' indicates the object pose accuracy after object viewpoint estimation (the first stage of VFM-6D). It is highly relevant to the density of the sampled reference images within the $SO(3)$ rotation space. As shown in Fig. 9, the baseline accuracy is relatively sensitive to the number of reference images. In contrast, VFM-6D is relatively stable with respect to the number of reference images.

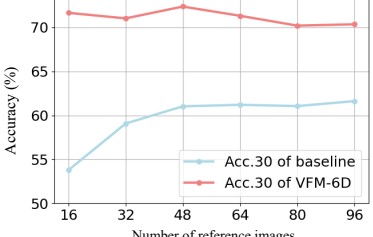

Figure 9: Results with different number of reference images.

## 4.5 VFM-6D in Open-world Scenarios

VFM-6D exhibits high generalization capability to handle various open-world scenarios. As shown in Fig. 10, based on the visual understanding capability of vision-language models (*e.g.*, GPT-4V [78]) and the generation capability of existing text-to-3D generative models (*e.g.*, Meshy [65]), VFM-6D can effectively exploit the generated category-level shape to parse precise object poses for novel

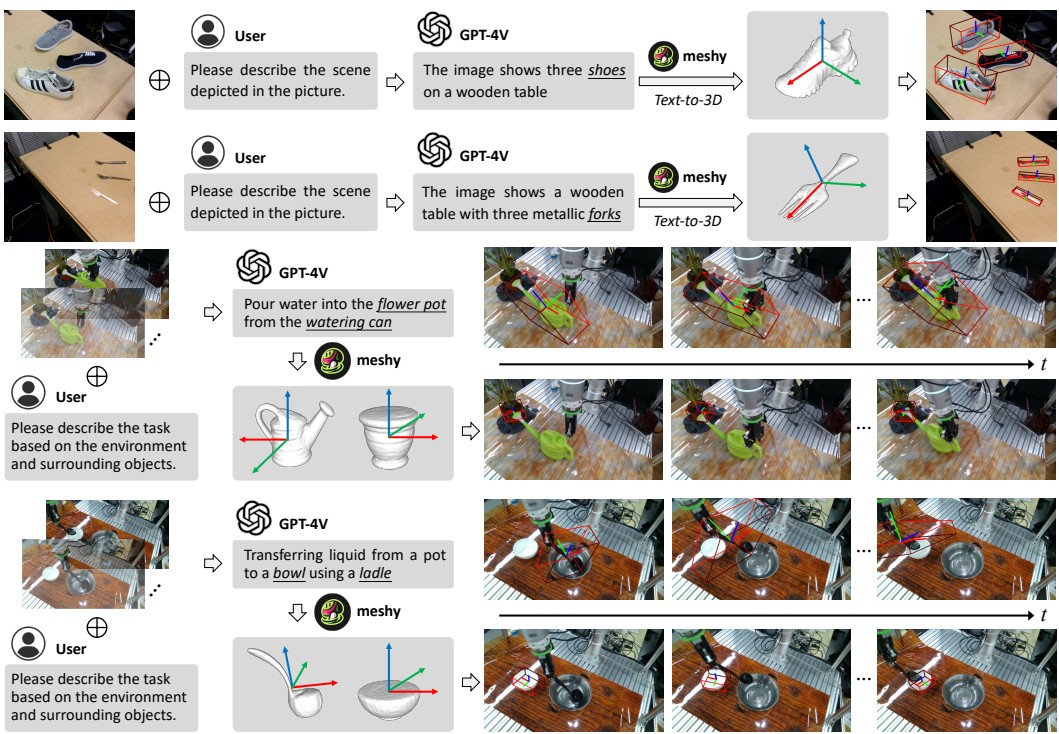

Figure 10: VFM-6D application in scenarios from D$^3$Fields (top) [76] and RH20T (bottom) [77].

objects that the agent encounter in open-world environments. This process does not require heavy instance-level data collection and labeling, and it is not confined to a pre-defined set of object categories. Moreover, VFM-6D exhibits the potential for parsing sequential object poses from the robotic manipulation video, further demonstrating its wide application scenarios for robot learning.

## 5  Conclusion and Discussion

In this paper, we introduce VFM-6D, a new approach for generalizable object pose estimation. VFM-6D incorporates a feature lifting module for viewpoint estimation and a shape representation module for robust shape matching. These modules, built on the vision foundation model, allow for joint estimation of object pose and size through correspondence-based optimization. VFM-6D has demonstrated superiority in evaluations, with high generalization for application in both instance-level and category-level pose estimation. Furthermore, its integration with vision-language and generative models allows effective handling of various open-world scenarios, showcasing its potential to boost 3D understanding capability of multimodal large language models.

VFM-6D has a few limitations that future work can improve. First, since the visual occlusion affects the pre-trained object representation from the vision foundation model, VFM-6D would also be affected by severe occlusions. Please refer to the appendix for the performance evaluation under different occlusion rates. Note that for the robot manipulation scenario, this problem could be addressed by active perception [46] and mobile manipulation [79] to find the occlusion-free viewpoint. Second, VFM-6D requires both RGB images and depth data, limiting its applicability to large-scale internet data, where depth information is often unavailable. In future work, we plan to explore integrating VFM-6D with depth foundation models [80, 81] to enhance generalizable object pose estimation. Additionally, RGB-based implicit object shape representations [82, 83] may provide a promising alternative to eliminate the need for depth data.

**Acknowledgements.** This project is supported in part by the Shenzhen Portion of Shenzhen-Hong Kong Science and Technology Innovation Cooperation Zone under HZQB-KCZYB-20200089, in part by the InnoHK Clusters of the Hong Kong SAR Government via the Hong Kong Centre for Logistics Robotics, and in part by the National Natural Science Foundation of China (Project No. 62322318). Pieter Abbeel holds concurrent appointments as a Professor at UC Berkeley and as an Amazon Scholar. This paper describes work performed at UC Berkeley and is not associated with Amazon.

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

# Appendix

In this appendix, we provide additional results and implementation details that are not included in the main paper due to the space limit: (A) More detailed information for the synthetic data that we used for VFM-6D training. (B) Implementation details for the development and training of VFM-6D. (C) Qualitative comparison results for object viewpoint estimation. (D) More detailed quantitative and qualitative results on the CO3D dataset. (E) Quantitative evaluation results on all 13 object instance of LINEMOD and the corresponding qualitative results. (F) Object pose estimation results under different occlusion rates. (G) Object pose estimation results with the depth map predicted by a depth estimation foundation model [81].

## A. Synthetic Data for VFM-6D Training

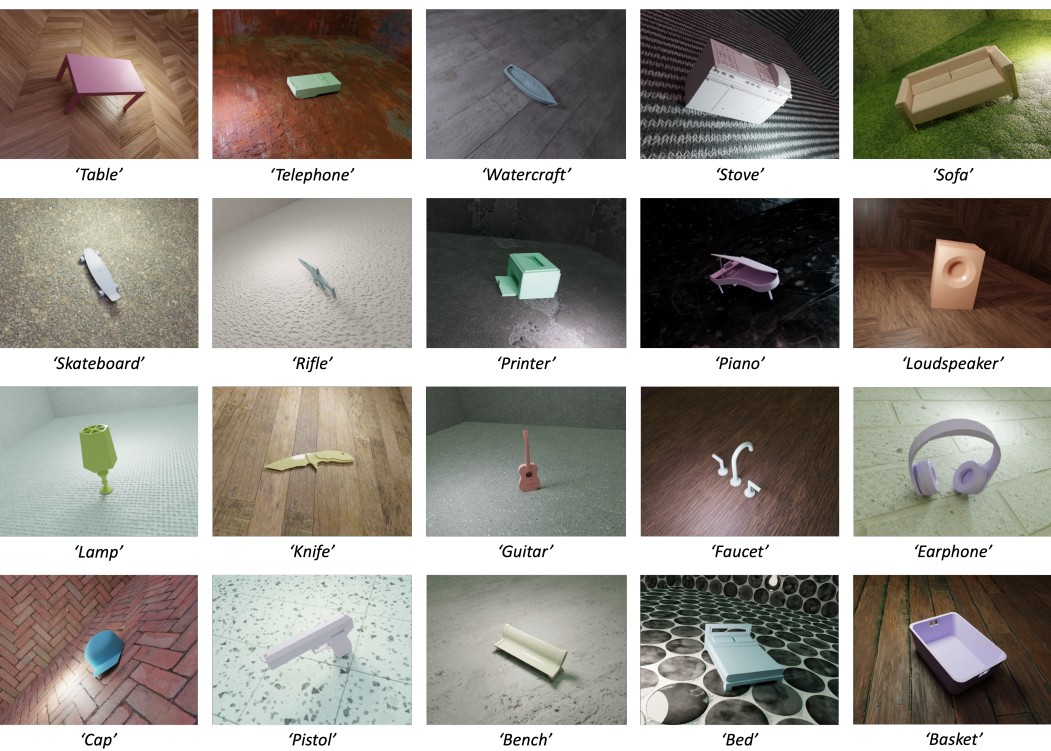

Figure 11: 20 object categories used for generating the synthetic data and their example images.

The synthetic dataset utilized for the training of VFM-6D, as depicted in Fig. 11, was generated using Blender. This dataset encompasses twenty frequently encountered object categories. For each of these categories, six distinct instances varying in shape were gathered from ShapeNet [60]. To enhance the diversity of the synthetic training data, we introduced randomness in several aspects during the Blender simulation. These include the pose and texture of the object, the texture of the background, the conditions of lighting, and the viewpoints of the camera. Similar to [59], two aspects of pose sampling are adopted to generate synthetic data with diverse object poses: (1) object on-surface sampling. The object would be placed upright onto a plane of the synthetic scene, and the in-plane position and orientation of the object would be randomly sampled. (2) camera pose sampling. The camera location is first sampled around the object using the "uniform_elevation" sampling strategy. Then, the camera rotation is further determined by a point of interest which is randomly sampled from the scene, plus a sampled camera in-plane rotation within a specified range. For each category, an object instance was randomly selected and used to synthesize a total of 10K RGBD images. Concurrently, the object mask and ground-truth object coordinate map were synthesized for each image. The synthetic training data comprises 200K RGBD images. It is important to note that the object categories employed for VFM-6D training do not overlap with any evaluation data utilized in the experiments.

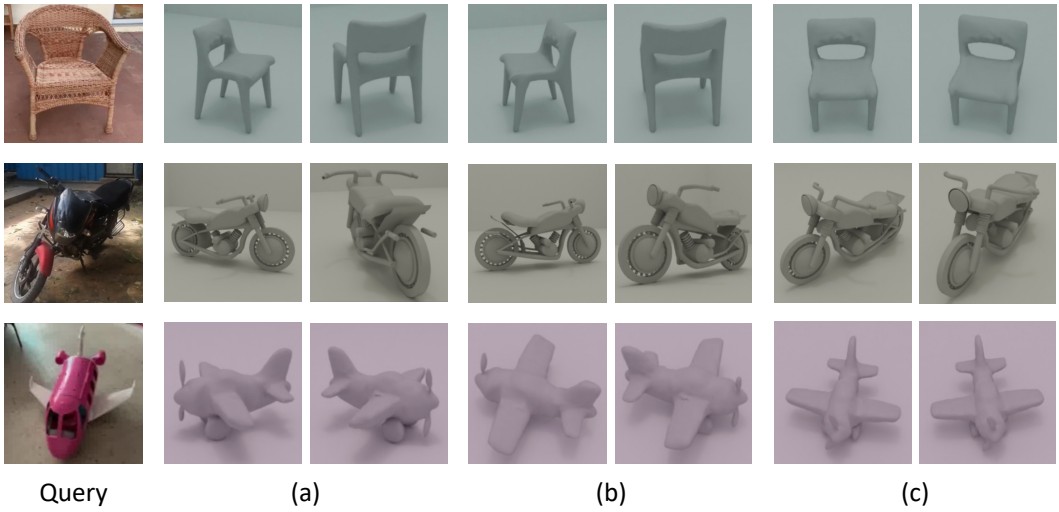

| Query | (a) | (b) | (c) |

Figure 12: Comparative results for object viewpoint estimation. We present the top-2 reference images found by different approaches.(a) results w/o the proposed foundation feature lifting module. (b) results of training from scratch + PCT. (c) results of proposed VFM-6D.

## B. Implementation Details

We developed VFM-6D based on DINOv2-s. Both query and reference images are cropped and resized to $224 \times 224$, and are tokenized with a patch size of $14$ before being fed into the vision foundation model backbone for feature extraction. We set $n = m = 1024$. It means that for both query and reference images, we randomly sample $1024$ object pixels based on the object mask. These pixels are further lifted into object point clouds based on the depth map and camera intrinsic parameters. For the proposed 2D-to-3D foundation feature lifting module, we set $N_1 = 1$, and the 3D position embedding layer adopts a 2-layer MLP. For the shape representation module, we set $N_2 = 4$. We set $\tau = 0.05$ for the pose-aware contrastive loss, and $\beta = 0.1$ for the smooth L1 loss. Unless specifically pointed out, we use $k = 64$ reference images for query-reference matching and object pose estimation. Please refer to Sec.F for results with different numbers of reference images. During training, we use Adam optimizer with a base learning rate of $1 \times 10^{-4}$ and halved every 3 epochs. The total training epoch is set to $15$ with a batch size of $8$. All experiments were conducted on a server with two NVIDIA-A40-48GB GPUs. Given the predicted NOCS coordinate map and object point cloud, we exploit Umeyama algorithm [84] to jointly optimize object pose and size parameters. RANSAC [85] was applied to VFM-6D and all other correspondence-based competing methods for robust estimation.

## C. Results for Object Viewpoint Estimation

Fig. 12 compares the object viewpoint estimation results of different approaches. More specifically, we compare VFM-6D with two approaches. The first one is VFM-6D without our proposed 2D-to-3D foundation feature lifting module, which estimate object viewpoint by foundation-feature-based template matching and corresponds to 'w/o feature lifting' or 'w/o both' in Table 4 of the main paper. The second one is the method that replaces the adopted vision foundation model with a point cloud Transformer and training the whole framework from scratch, which corresponds to 'training from scratch+PCT' in Fig. 7 of the main paper. We provide comparative results to facilitate the understanding of their performance drop when compared with VFM-6D. As can be observed in Fig. 12, foundation-feature-based template matching suffers from the shape variation between query and reference images and associates the query image with very distant references. Although PCT boosts the performance in the setting of 'training from scratch', it still suffer a limited generalization capability and would fail to match the query image with its correct reference. In contrast, VFM-6D is much more robust in object viewpoint estimation.

# D. More Results on CO3D

As shown in Table 5, we present detailed comparative results for all 20 object categories. We show qualitative results of VFM-6D in Fig. 13. It can be observed that our proposed VFM-6D can precisely associate the query image with its close reference image. Moreover, VFM-6D can robustly match the query object shape with the reference one, and can estimate object pose and size accurately for a wide range of objects in different textures and shapes.

Table 5: Category-level object pose estimation results on 20 unseen categories of CO3D dataset. We report per-category and average Acc.$15°$ / Acc.$30°$ for quantitative evaluation.

| Method | Motorcycle | Backpack | Bicycle | Teddybear | Book | Car | Chair |
|---|---|---|---|---|---|---|---|
| LoFTR [71] | 4.2 / 10.9 | 1.1 / 2.8 | 2.2 / 7.8 | 12.5 / 18.9 | 0.1 / 1.3 | 30.0 / 36.2 | 8.9 / 15.2 |
| LightGlue [72] | 4.2 / 11.8 | 0.5 / 4.9 | 14.4 / 27.0 | 8.2 / 19.5 | 1.0 / 3.0 | 7.5 / 13.8 | 3.5 / 8.2 |
| GeDi [73] | 4.3 / 11.5 | 0.5 / 2.7 | 5.6 / 17.6 | 9.4 / 25.7 | 0.6 / 1.7 | 0.5 / 3.0 | 10.5 / 23.0 |
| ZSP [52] | 25.0 / 49.2 | 4.4 / 13.0 | 27.0 / 49.3 | 23.7 / 43.4 | 1.5 / 3.2 | 61.8 / 69.3 | 30.0 / 41.6 |
| Ours | 56.4 / 76.3 | 30.6 / 47.4 | 46.5 / 59.2 | 25.2 / 54.4 | 41.9 / 43.5 | 55.6 / 74.4 | 72.1 / 86.8 |

| Method | Handbag | Hydrant | Keyboard | Mouse | Toaster | Hairdryer | Laptop |
|---|---|---|---|---|---|---|---|
| LoFTR [71] | 5.8 / 12.6 | 17.7 / 39.1 | 7.2 / 12.7 | 3.4 / 5.2 | 7.2 / 12.9 | 4.6 / 8.4 | 40.3 / 47.0 |
| LightGlue [72] | 10.8 / 28.0 | 8.0 / 22.1 | 8.0 / 15.8 | 0.4 / 3.2 | 5.7 / 9.7 | 0.5 / 3.7 | 26.2 / 30.6 |
| GeDi [73] | 5.7 / 15.3 | 19.6 / 52.8 | 6.4 / 15.0 | 1.9 / 10.5 | 0.2 / 0.8 | 6.4 / 16.7 | 7.7 / 13.4 |
| ZSP [52] | 23.6 / 49.9 | 38.1 / 88.9 | 21.7 / 29.8 | 18.1 / 40.7 | 26.1 / 36.0 | 33.9 / 63.5 | 80.7 / 86.4 |
| Ours | 75.5 / 89.5 | 35.4 / 91.6 | 57.1 / 57.1 | 38.3 / 57.3 | 44.2 / 47.7 | 63.0 / 85.2 | 96.8 / 97.5 |

| Method | Remote | Toilet | Toybus | Toyplane | Toytrain | Toytruck | Average |
|---|---|---|---|---|---|---|---|
| LoFTR [71] | 4.8 / 8.9 | 2.4 / 8.5 | 2.1 / 7.1 | 6.1 / 2.1 | 5.9 / 11.1 | 2.0 / 4.9 | 8.4 / 13.7 |
| LightGlue [72] | 2.9 / 4.6 | 0.3 / 1.5 | 3.9 / 13.1 | 3.5 / 9.5 | 6.2 / 15.4 | 2.5 / 8.3 | 5.9 / 12.7 |
| GeDi [73] | 4.1 / 10.6 | 1.1 / 4.3 | 5.0 / 12.4 | 6.0 / 13.8 | 7.6 / 12.2 | 2.1 / 4.3 | 5.3 / 13.4 |
| ZSP [52] | 13.2 / 23.6 | 8.4 / 25.0 | 14.9 / 43.3 | 34.9 / 49.5 | 41.2 / 57.2 | 14.2 / 30.5 | 27.1 / 44.7 |
| Ours | 33.3 / 36.8 | 47.1 / 75.1 | 24.8 / 53.4 | 55.1 / 66.6 | 61.9 / 80.2 | 43.4 / 67.8 | **50.2 / 67.4** |

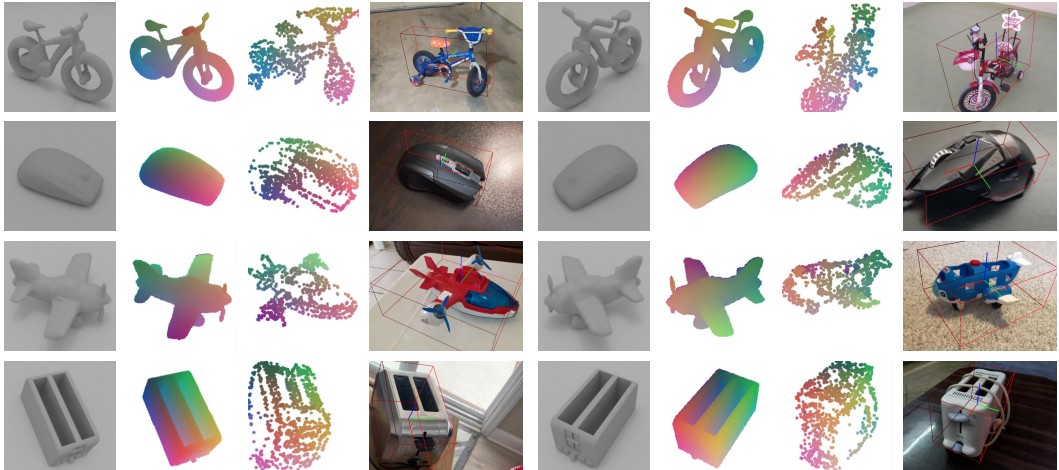

Figure 13: Qualitative results of VFM-6D on '*bicycle*', '*mouse*', '*toyplane*', and '*toaster*' of CO3D. From left to right, the 1st and 5th columns are matched reference images, the 2nd and 6th columns are coordinate maps of reference images, the 3rd and 7th columns are predicted coordinate maps for the query image, and the 4th and 8th columns are object pose estimation results.

# E. More Results on LINEMOD

As shown in Table 6, we present detailed comparative results for all 13 object instances in LINEMOD. As can be observed, VFM-6D is also applicable to the instance-level setting, if we take each

object instance as one object category. VFM-6D outperforms two zero-shot baselines and achieves comparable performance to the few-shot baseline. Fig. 14 depicts qualitative results on LINEMOD.

Table 6: Instance-level object pose estimation results measured by ADD-0.1d on LINEMOD dataset. We report average score over all instances and per-instance score for an illustrative subset of instances. Please refer to the appendix for full per-instance results.

| Method | ape | bench. | camera | can | cat | driller | duck |
|---|---|---|---|---|---|---|---|
| LatentFusion [68] | 88.0 | 92.4 | 74.4 | 88.8 | 94.5 | 91.7 | 68.1 |
| OSOP [69] | 86.1 | 94.6 | 69.3 | 80.4 | 88.0 | 79.5 | 92.7 |
| FS6D [70] | 78.0 | 88.5 | 91.0 | 89.5 | 97.5 | 92.0 | 75.5 |
| Ours | 88.5 | 91.9 | 97.2 | 74.9 | 96.4 | 85.1 | 88.0 |

| Method | eggbox | glue | holep. | iron | lamp | phone | Avg. |
|---|---|---|---|---|---|---|---|
| LatentFusion [68] | 96.3 | 94.5 | 82.1 | 74.6 | 94.7 | 91.5 | 87.1 |
| OSOP [69] | 98.2 | 69.5 | 92.7 | 99.4 | 48.5 | 66.1 | 82.0 |
| FS6D [70] | 99.5 | 99.5 | 96.0 | 87.5 | 97.0 | 97.5 | 91.5 |
| Ours | 97.1 | 99.1 | 91.2 | 97.5 | 77.0 | 89.8 | 90.3 |

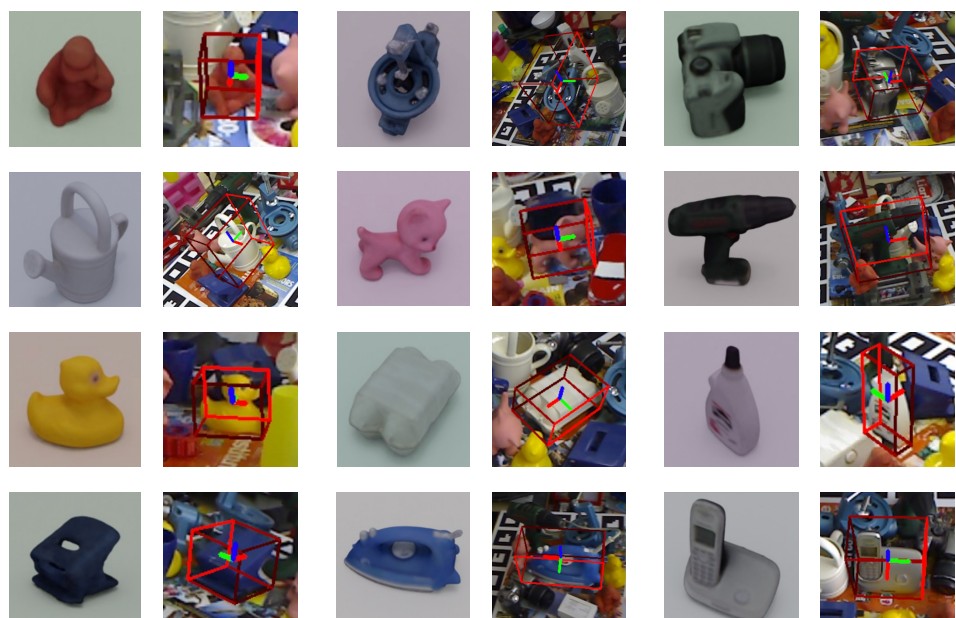

Figure 14: Qualitative results of VFM-6D on LINEMOD. For each object instance, the left image is the reference image found by VFM-6D. The right image is the corresponding query image, on which the object pose estimation results are overlayed.

## F. Object Pose Estimation Results under Occlusions

To investigate this problem, we first evaluated VFM-6D under different levels of occlusion on the LINEMOD-Occlusion dataset [86]. Moreover, we also conducted more comprehensive evaluations on the CO3D dataset. Since scenes from CO3D are generally occlusion-free, we randomly masked out different percentages of image regions to mimic the effect of object occlusion. Table 7 and Table 8 report the quantitative evaluation results. These results show that VFM-6D is relatively robust to small and moderate occlusion rates, and would suffer an apparent performance drop when facing severe occlusions larger than 60% occlusion rate.

Table 7: Object pose estimation results on LINEMOD and LINEMOD-Occlusion datasets.

| | No occlusion | <30% | 30%-60% | >60% |
|---|---|---|---|---|
| ADD-0.1d | 90.3 | 87.4 | 69.4 | 32.4 |

Table 8: Object pose estimation results on 5 representative object categories of the CO3D dataset.

| | No occlusion | <30% | 30%-60% | >60% |
|---|---|---|---|---|
| Motorcycle | 56.4 / 76.3 | 54.9 / 70.1 | 43.3 / 64.1 | 32.3 / 51.0 |
| Chair | 72.1 / 86.8 | 68.2 / 82.0 | 55.3 / 74.1 | 38.4 / 57.7 |
| Bicycle | 46.5 / 59.2 | 45.9 / 60.5 | 42.9 / 61.6 | 39.0 / 60.2 |
| ToyPlane | 55.1 / 66.6 | 53.0 / 64.3 | 44.8 / 56.4 | 45.0 / 55.5 |
| ToyTrain | 61.9 / 80.2 | 61.6 / 77.1 | 46.3 / 76.6 | 40.9 / 71.1 |

## G. Object Pose Estimation Results with Depth Anything

To investigate the performance of VFM-6D in some RGB-only scenarios, we input the RGB image into the pre-trained depth-anything [81] model to recover the corresponding metric depth. Then, we input the RGB image and the recovered depth into our VFM-6D for object pose estimation. Without any additional training, Table 9 reports the evaluation results on the CO3D dataset. As can be observed, VFM-6D can effectively leverage the depth map estimated from the RGB image for generalizable object pose estimation. Note that in the context of category-level object pose estimation, we usually jointly estimate object size and pose parameters. Based on the estimated metric depth map, our method can recover object rotation precisely and can recover object size and translation up to a global scale factor. The RGB-only variant achieves comparable rotation accuracy on average when compared with the original RGBD-based performance. Please also refer to Fig. 15 for detailed qualitative pose prediction results.

Table 9: Object pose estimation results with Depth-anything model on the CO3D dataset.

| Method | Motorcycle | Backpack | Bicycle | Teddybear | Book | Car | Chair |
|---|---|---|---|---|---|---|---|
| RGB-D | 56.4 / 76.3 | 30.6 / 47.4 | 46.5 / 59.2 | 25.2 / 54.4 | 41.9 / 43.5 | 55.6 / 74.4 | 72.1 / 86.8 |
| RGB-only | 49.7 / 68.4 | 26.4 / 48.5 | 44.6 / 52.5 | 29.0 / 56.9 | 38.5 / 39.7 | 43.1 / 63.3 | 81.5 / 94.4 |

| Method | Handbag | Hydrant | Keyboard | Mouse | Toaster | Hairdryer | Laptop |
|---|---|---|---|---|---|---|---|
| RGB-D | 75.5 / 89.5 | 35.4 / 91.6 | 57.1 / 57.1 | 38.3 / 57.3 | 44.2 / 47.7 | 63.0 / 85.2 | 96.8 / 97.5 |
| RGB-only | 40.2 / 77.3 | 61.2 / 98.9 | 29.4 / 48.8 | 23.9 / 52.5 | 39.1 / 43.9 | 52.8 / 70.6 | 85.2 / 98.2 |

| Method | Remote | Toilet | Toybus | Toyplane | Toytrain | Toytruck | Average |
|---|---|---|---|---|---|---|---|
| RGB-D | 33.3 / 36.8 | 47.1 / 75.1 | 24.8 / 53.4 | 55.1 / 66.6 | 61.9 / 80.2 | 43.4 / 67.8 | **50.2 / 67.4** |
| RGB-only | 22.5 / 39.2 | 53.5 / 80.1 | 21.3 / 46.9 | 46.0 / 63.2 | 51.9 / 79.5 | 37.7 / 62.0 | 43.9 / 64.2 |

## H. Discussion on Broader Impacts

In this work, we present a generalizable object pose estimation framework VFM-6D for instance-level unseen object pose estimation and category-level object pose estimation for novel categories. VFM-6D is crucial for the development of safe and reliable robotic manipulation processes. By accurately parsing the novel object pose in the scene, VFM-6D enables robots to perceive and interact with objects in their environment with increased accuracy and confidence. This capability empowers robots to handle delicate or complex novel objects, reducing the risk of damage or errors during manipulation tasks. The precise and adaptable pose estimation provided by VFM-6D facilitates informed decision making, improves safety measures, and supports sustainable practices.

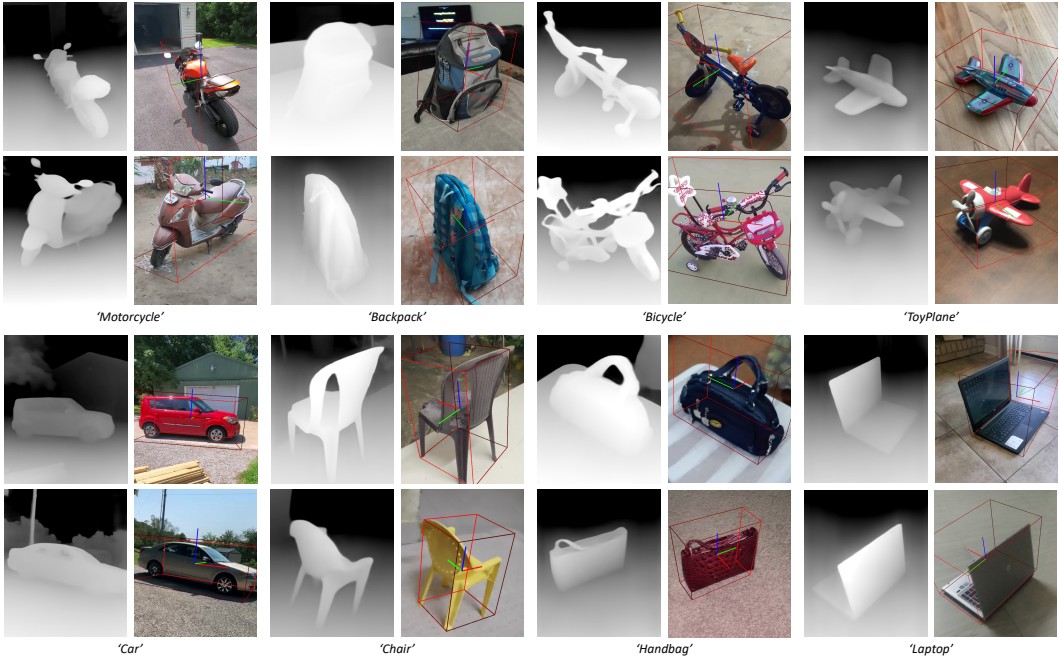

'Motorcycle'      'Backpack'      'Bicycle'      'ToyPlane'

'Car'      'Chair'      'Handbag'      'Laptop'

Figure 15: Qualitative results of VFM-6D + Depth-anything model on the CO3D dataset. For each pair of results, the left image depicts the predicted depth map and the right image visualizes the predicted object pose.

