# OpenReview forum: "Vision Foundation Model Enables Generalizable Object Pose Estimation"
_NeurIPS.cc/2024/Conference — NeurIPS 2024 poster_

### Official Review · Reviewer_gYMD · 2024-07-04

**Soundness:** 2
**Presentation:** 3
**Contribution:** 2
**Rating:** 4
**Confidence:** 4

**Summary:**

This paper introduces VFM-6D, a two-stage RGBD-based method for generalizable object pose estimation. Given a set of reference images depicting objects of the arbitrary category, the proposed method first estimates a viewpoint using image matching, then based on the NOCS map of this matched reference image, it estimates the NOCS map of the query image, which allows for a more accurate re-estimation of the 6D pose and 3D size of the input objects. The paper experiments on novel categories of category-level object pose benchmarks such as Wild6D, CO3D, and unseen instance object pose estimation LINEMOD, showing state-of-the-art results on these benchmarks.

**Strengths:**

- S1: The paper presents a generalizable method, VFM-6D, that can be applied to both instance-level unseen object pose estimation and category-level object pose estimation for novel categories.

- S2: The paper is well-structured and easy to follow.

- S3: The experiments demonstrate that VFM-6D achieves state-of-the-art results on several benchmarks.

**Weaknesses:**

- W1: The main contribution of the paper, category-level object pose estimation for novel categories when reference and query images are very different, is not well supported. There is no analysis to back up this claim, and experiments either do not show how the reference and query objects differ (Wild6D, CO3D) or the reference and query objects are the same (LINEMOD).

- W2: The quantitative results in Table 1, 2, and 3 are unclear. Some methods, such as PoseContrast, LoFTR, and LightGlue, are RGB-based, while VFM-6D uses both RGB and depth as input, which contains more information for estimating 6D pose. It would be helpful if the authors clearly stated the inputs used in each method for a fair comparison. Additionally, it is unclear whether these results come from re-training the baseline on the same training set or using available pre-trained models.

- W3: The paper does not clearly explain why a two-stage approach is necessary, as the first stage can already provide the 6D pose of the object via the pose of the nearest reference, which can serve as the pose prediction. If this is for pose refinement purposes, it would be well-motivated and much clearer if the authors showed the results of the first stage.

- W4: (Minor) The paper lacks implementation details, such as how 64 reference images are sampled (only cover out-of-plane rotation or both out-of-plane rotation and in-plane rotation), which tokens are being used in foundation features (class token or patch token), how the figure 1 is made and similarity is normalized.

**Questions:**

All my questions are related to the weaknesses mentioned above:

- Q1: Can the proposed method work well on unseen categories when the geometry/texture of reference and query objects are different?

- Q2: Did the authors experiment with GPT-4V and Text-to-3D for generating 3D models and estimating 6D pose as shown in Figure 2? If yes, which models were used?

- Q3: How 64 reference views are sampled?

- Q4: Which tokens are used in foundation features?

- Q5: In Figure 1, did the authors use the same constant for normalizing the similarity between two methods, and is the similarity too low, between 0.006 and 0.0014 but not 0 and 1? Where is the GT view ID?

- Q6: Please clearly mention how the authors obtained the results for the baseline, and clearly mention the input used in each baseline.

**Limitations:**

Yes, the authors mentioned the limitations which is requiring the depth image as the input.

---

> ### Author Rebuttal · Authors · 2024-08-07
>
> >**Q1: Can the proposed method work well on unseen categories when the geometry/texture of reference and query objects are different?**
>
> **A1:** Thanks for the comment. We suppose that our presented experiments on Wild6D and CO3D datasets could address this concern. For Wild6D evaluation, we have tested our method on 162 different object instances in 5 unseen object categories. For CO3D evaluation, we have tested our method on 200 different object instances in 20 unseen object categories. In Fig.2 of the PDF file uploaded in the global response, we present some examples of reference and query objects used in Wild6D and CO3D evaluations.
> - In terms of the texture difference, the reference object that we used is untextured, while the query objects in various testing scenarios usually have colorful and diverse textures. The texture difference between reference and query objects is significant.
> - In terms of the geometry difference, since the precise CAD model for each object instance in Wild6D and CO3D is not available, we are not able to measure the geometry difference quantitatively. However, as depicted in Fig.2 of the uploaded PDF file, the geometry difference between reference and query objects is rather apparent. For example, the reference ‘mug’ can be used to estimate poses for a variety of mugs with different rim shapes and handle styles. Moreover, the reference ‘chair’ with four legs and no armrests can be used to estimate poses for various chairs with different leg structures and chairs with or without armrests.
>
> We suppose that these results can demonstrate that our method can work well on unseen categories when the geometry/texture of reference and query objects are different.
>
> >**Q2: Why is a two-stage approach necessary?**
>
> **A2:** Thanks for the comment. In our ablation study, we have validated our designs on the first and second stages. Table 4 of our original paper demonstrates the significant performance improvement brought by each individual module. If we only use the first stage, the final pose accuracy would be sensitive to the number/pose distribution of sampled reference images. To explore this problem, we evaluated our method with different numbers of reference images. These result has been presented in Sec.F of the paper appendix. As shown in Fig.14 of the appendix, the second stage makes the final pose accuracy become robust to the selection of reference images. Moreover, Fig.14 also comprehensively compared the pose accuracy after the first stage and the second stage. The second stage can consistently improve the performance. These results demonstrate the advantage and necessity of the two-stage design.
>
> >**Q3: The specific required inputs and evaluation settings used for each baseline.**
>
> **A3:** Thanks for your valuable suggestion. To clarify, we list the involved baseline methods one by one:
> - SPD, SGPA, DualPoseNet, and GPV-Pose: RGBD-based methods. We re-trained their models and evaluated them on Wild6D.
>
> - PoseContrast: Use RGB images during object viewpoint estimation, and further leverage depth map to estimate complete object pose. We re-trained the PoseContrast model with our constructed synthetic dataset and then evaluated it on Wild6D.
>
> - ZSP: RGBD-based baseline. Does not involve any training module and we directly evaluated it based on its official source code.
>
> - LoFTR and LightGlue: As mentioned in L299-300 of the original paper, we need to use both RGB images and depth maps for category-level object pose estimation. We exploited their officially provided pre-trained models during experiments.
>
> - GeDi: Point-cloud-based method. As a generalizable local deep descriptor for point cloud registration, we exploited its robust model pre-trained on 3DMatch during experiments.
>
> We are sorry for the possible confusion caused. We will elaborate on the description for all baseline approaches in our revised version.
>
> >**Q4: Did the authors experiment with GPT-4V and Text-to-3D for generating 3D models and estimating 6D poses as shown in Figure 2? Which models were used?**
>
> **A4:** Thanks for your interest in our GPT-4V + Text-to-3D setting. We mainly tested this setting in practical open-world scenarios from the D3-Field dataset and RH20T dataset. Figure 9 of the main paper and Figure 15 of the paper appendix present the experiment results on various unseen object categories, including ‘shoes’, ‘flower pot’, ‘watering can’, ‘forks’, and ‘ladle’. For experiments on Wild6D and CO3D, we leveraged the collected shape templates for evaluation.
>
> >**Q5: How are the 64 reference views sampled?**
>
> **A5:** Thanks for the comments. The procedure of preparing shape templates can be divided into the following three steps: (1) Normalize the input object CAD model to have a diameter of 1; (2) 64 camera viewpoints are sampled from a sphere centered on the normalized object CAD model; (3) Rendering the corresponding RGB-D template image at each viewpoint with Blender. At each camera viewpoint, we also randomly sample the camera in-plane rotation within a specified rotation range (i.e., [$-10^\circ$, $10^\circ$]).
>
> >**Q6: Which tokens are used in foundation features?**
>
> **A6:** Thanks for the comment. For CLIP and MVP foundation models, we used their patch tokens. For DINO-v1 and DINO-v2 models, we developed our framework based on their ‘key’ tokens. We will incorporate these implementation details during revision.
>
> >**Q7: Clarify Figure 1. How is the similarity score normalized?**
>
> **A7:** Thanks for your careful checking. Given one query image and $N$ reference images, we would obtain $N$ raw similarity scores. We further normalize the similarity score via softmax. Note that a global scale factor of 0.5 is applied to the normalized score for the purpose of visualization, which does not affect the similarity distribution over reference views. Sorry for any confusion caused. We will further revise Figure 1 and its corresponding caption to improve its clarity.

---

> > ### Comment · Reviewer_gYMD · 2024-08-11
> > **Raising score**
> >
> > I would like to thank the authors for very detailed replies (both for me and other reviewers). The rebuttal addresses all of my concerns, and I am willing to raise my scores.

---

> ### Author Response · Authors · 2024-08-11
> **Thank You!**
>
> Dear Reviewer gYMD,
>
>
> Thank you for your feedback on our rebuttal. We are very glad that our response has fully addressed your earlier questions. Much appreciate your kind support for our work. We are grateful for your willingness to raise the score. Thanks a lot!
>
>
> Sincerely,
>
> Authors

---

### Official Review · Reviewer_heqf · 2024-07-08

**Soundness:** 3
**Presentation:** 3
**Contribution:** 3
**Rating:** 7
**Confidence:** 5

**Summary:**

The paper presents VFM-6D, a new framework for generalizable object pose estimation. VFM-6D integrates a 2D-to-3D feature lifting module and a shape-matching module, both of which utilize pre-trained vision foundation models to enhance object representation and matching accuracy. In open-set robotic manipulation scenarios, the authors use GPT-4v to generate mesh information from the prompts. The model is trained on synthetic data and demonstrates promising generalization capabilities in various real-world scenarios, as evidenced by evaluations on Wild6D and CO3D datasets.

**Strengths:**

•  This paper tackles a challenging but important problem in generalizable object pose estimation. Unlike previous approaches which require either a 3D mesh or reference images of the same object instance, the presented method can predict the 6D object pose given an RGBD image, utilizing category-level reference information.

•  Since the appearance and shape of the reference instance could be different from those of the query instance, the pose estimation is quite difficult. The authors propose to handle this problem by performing template matching and registration, which is technically sound.

•  The experimental results are promising. The method is trained on synthetic images, but tested on real unseen images. It outperforms some previous approaches on Wild6D and CO3D datasets.

•  The authors leverage GPT-4v to generate reference information, which may facilitate the applications in real-world scenarios.

**Weaknesses:**

•  The method relies heavily on strong prior information, such as depth maps and NOCS maps. This requirement could limit its applicability in real-world scenarios.

•  Some important details are missing. Specifically, how to perform the object shape focalization is not introduced; how to obtain the NOCS map for a reference image is unclear.

•  The idea of utilizing foundation models is not very impressive. There have been some approaches that use foundation models for generalization object pose estimation. I would recommend rewriting the introduction, highlighting the importance and the novelty of the category-level setting.

•  For query-reference image matching, some comparisons are missing. [1][2][3] use a patch-level template matching strategy for image matching, which does not rely on depth information but achieves good generalization ability.

[1] Nguyen, Van Nguyen, et al. "Templates for 3d object pose estimation revisited: Generalization to new objects and robustness to occlusions." Proceedings of the IEEE/CVF conference on computer vision and pattern recognition. 2022.
[2] Zhao, Chen, Yinlin Hu, and Mathieu Salzmann. "Fusing local similarities for retrieval-based 3d orientation estimation of unseen objects." European Conference on Computer Vision. Cham: Springer Nature Switzerland, 2022.
[3] Örnek, Evin Pınar, et al. "Foundpose: Unseen object pose estimation with foundation features." arXiv preprint arXiv:2311.18809 (2023).

•  The experiment on CO3D is somewhat unfair. All evaluated methods are based on a feature-matching technique. The presented method is trained on ShapeNet and tested on CO3D. Both of them are object-centric. However, other methods such as LoFTR and LightGlue are trained on scene-centric images, which may be not able to generalize to object-centric scenarios without finetuning.

**Questions:**

Please refer to my comments in "Weaknesses"

**Limitations:**

The limitations have been thoroughly discussed by the authors.

---

> ### Author Rebuttal · Authors · 2024-08-07
>
> >**Q1: The method relies on depth maps and NOCS maps, which could limit its applicability in real-world scenarios.**
>
> **A1:** Thanks for your comment. To address your concern, we explore the possibility of RGB-only pose prediction for our method. Due to the character limit, please refer to **Q1** in the global response for detailed evaluation results.
>
> The NOCS map of a reference image could be directly computed based on the GT pose of the reference image. It does not require any other additional information. Please refer to our response to your Q3 for details of computing the NOCS map. In this regard, the use of the NOCS map would not be a limitation of our method in practical applications.
>
> >**Q2: How to perform the object shape focalization?**
>
> **A2:** Thanks for your comment. The focalization step takes as input the object point cloud in the camera frame and the initial object orientation estimated during the query-reference image matching stage. It aims to transform the object point cloud into a relatively canonical and normalized space to facilitate the subsequent object shape representation. Formally, the focalization step can be formulated as $\mathcal{X}'=\mathbf{R}^{-1}\times\frac{\mathcal{X}-\bar{\mathcal{X}}}{\max{\|\|\mathcal{X}-\bar{\mathcal{X}}\|\|}}$, where $\mathcal{X}$ denotes the original object point cloud in camera frame, $\bar{\mathcal{X}}$ denotes point cloud center, and $\mathbf{R}$ is the initial object orientation estimated during the query-reference image matching stage. We will elaborate on this part more clearly in the revised version.
>
> >**Q3: How to obtain the NOCS map for a reference image?**
>
> **A3:** Thanks for pointing it out. Each pixel in the NOCS map indicates a 3D coordinate value in the normalized object frame. For each reference image, the object scale $s$ and the ground-truth object pose $[\mathbf{R}|\mathbf{t}]$ should be known. For each pixel in the camera frame, we are able to transform its 3D coordinate into the object frame based on $[\mathbf{R}|\mathbf{t}]$ and then normalize the coordinate value with $s$ to obtain the corresponding coordinate value in the NOCS map. We feel sorry for the possible confusion and will elaborate on these details in the revised version.
>
> >**Q4: Recommend rephrasing the introduction to highlight the importance and novelty of the category-level setting.**
>
> **A4:** Thanks for the constructive suggestion. We are glad to see that the reviewer appreciated the importance of the problem that our paper targets and the novelty of our method from the category-level perceptive. We suppose that our method could be distinguished from other existing foundation-model-based works from the following aspects: (1) We show that by using our proposed adaptable framework, we can effectively enhance the 3D representation capability of the pre-trained 2D foundation model to handle intra-class shape variations of different objects for challenging category-level object pose estimation. (2) We demonstrate that the pre-trained foundation model can be effectively adapted to the task of category-level object pose estimation with cost-effective synthetic data and also keeps high robustness in handling novel object categories. We would follow your suggestion to rephrase our paper to highlight our contributions from the category-level perspective.
>
> >**Q5: For query-reference image matching, comparison with patch-level template matching strategy [1][2][3] is missing.**
>
> **A5:** Thanks for the comment. The first patch-level template matching method [1] you suggested is Ref [71] in our original paper (cf. L322 on page 8). We have compared this template-matching strategy with our proposed feature lifting module in our ablation study. Table 4 of the original paper reports quantitative evaluation results (i.e., ‘w/o feature lifting’ vs. Ours) and Fig.11 of the appendix presents qualitative evaluation results (i.e., column (a) vs. column (c)). We suppose that these results can demonstrate the advantage of our method over the patch-level template matching strategy. We will rephrase our presentations to present this part more clearly.
>
> >**Q6: LoFTR and LightGlue are trained on scene-centric images, which may be not able to generalize to object-centric scenarios without finetuning.**
>
> **A6:** We appreciate you bringing this up. We have followed your suggestion and fine-tuned LightGlue on object-centric images. Specifically, we leveraged the 120 object models we had collected from ShapeNet to synthesize object-centric image pairs for fine-tuning LightGlue. For each object instance, we sampled image pairs by rotating the camera within a range of 30 degrees. We generated 200 image pairs for each object instance, resulting in a total of 24K image pairs used to fine-tune LightGlue. The table below presents the evaluation results of the fine-tuned LightGlue on the CO3D dataset.
> |        | Pre-trained LightGlue | Finetuned LightGlue | Ours |
> | :------: | :---------------------: | :-------------------: | :----: |
> | Acc.$15^\circ$ | 5.9                   | 9.8                 | 50.2 |
> | Acc.$30^\circ$ | 12.7                  | 15.4                | 67.4 |
>
> We found that for the task of category-level object pose estimation, the improvement from the object-centric fine-tuning was relatively limited. We suppose that the limitation of these feature-matching techniques is that they focus too heavily on low-level object features, and lack a sufficiently informative representation to perceive the object's shape and semantics, which makes them struggle with the challenging problem of category-level object pose estimation.
>
> [1] Nguyen et al. Templates for 3D Object Pose Estimation Revisited: Generalization to New Objects and Robustness to Occlusions. CVPR 2022.
>
> [2] Zhao et al. Fusing Local Similarities for Retrieval-based 3D Orientation Estimation of Unseen Objects. ECCV 2022.
>
> [3] Örnek et al. FoundPose: Unseen Object Pose Estimation with Foundation Features. ECCV 2024.

---

> > ### Comment · Reviewer_heqf · 2024-08-10
> >
> > I sincerely thank the authors for the rebuttal. Some of my concerns have been addressed. However, the experiment about depth maps and the explanation of the reference NOCS map are questionable to me.
> >
> > In that experiment, the authors use the depth anything model to predict metric depth from an RGB image. The method is still based on RGB-D input, not only RGB images. The metric depth prediction is much more challenging than relative depth prediction, and it is hard to generalize to diverse scenarios. The reliance on depth information is still a limitation of this method. Besides, it is confusing that the method with the predicted depth performs much better in some cases than that with the GT depth.
> >
> > The answer regarding the reference NOCS map seems wrong. The 2D-3D correspondences cannot be generated without depth.

---

> > > ### Author Response · Authors · 2024-08-12
> > > **Thank You and Our Further Response**
> > >
> > > Dear Reviewer heqf:
> > >
> > > Thank you for your kind feedback on our rebuttal.
> > >
> > > - We appreciate your further clarification on your question, which makes us more clearly understand your major concern regarding the depth map. We would like to gently clarify that our problem setting is different from what you indicate. Based on our understanding, you meant inputs are RGB images. In our setting, we used RGB-D data as the input, which is what has been usually assumed in the literature of category-level object 6D pose estimation. Upon re-interpreting your comments, we become aware that your problem setting is in fact a more challenging but meaningful one. We have newly searched the literature and interestingly found several very recent works investigating it. These methods [1,2,3,4] exploit RGB images to implicitly learn an object shape representation for object pose estimation, which are distinct from our framework. However, their current performance is still limited due to its challenging setting. Despite it being out of the scope of our this particular work, we sincerely thank you for your inspiring question, and we have re-considered the possibility of adapting our method to fit your more challenging problem setting. We conjecture that by changing our method’s depth-based explicit object shape modeling to RGB-based implicit shape modeling with latent feature volume[1,2] or implicit neural radiance fields[3,4], it is promising to effectively address the problem of object shape representation without any reliance on depth. This idea also aligns with the recent literature which addresses the same problem of RGB category-level object pose estimation as you mentioned. We very much appreciate your inspiring comments and will further extend our method to fit the more interesting setting, and thus making the pose estimation method even more generalizable to diverse scenarios in the future. We are also happy to include relevant discussions on this interesting point in our final version.
> > >
> > >   In addition, we would like to add clarifications to your question regarding the pose performance with predicted depth maps. Thank you for raising this interesting observation. After double-checking the experiments with the depth-anything model, the pose accuracy with the predicted depth map was found to be clearly higher than the one with the original CO3D depth map in 2 out of the 20 total categories (*i.e.*, chair and hydrant). After a meticulous review of the corresponding depth maps, we hypothesize that this may be due to the fact that, in comparison to the original CO3D depth, the predicted depth might provide a more comprehensive representation for certain chairs with backrests and handles that possess an openwork, perforated, or lattice-like structure. Furthermore, the predicted depth could potentially offer a smoother depiction for metal hydrants. Consequently, the predicted depth may be more capable of capturing the unique shape of the object for the category of 'chair' and 'hydrant', thereby resulting in improved pose accuracy. We are grateful for your insightful query and hope that this explanation offers a satisfactory response.
> > >
> > > - We apologize if we initially misunderstood your concern pertaining to the NOCS map. Yes, in our RGBD-based setting, we require depth information to establish correspondences between the object frame and the camera frame for pose optimization. Due to the inherent size ambiguity of objects in category-level object pose estimation, we are unable to directly utilize the 2D-3D correspondences and solve the pose parameters via the Perspective-n-Point algorithm. Instead, it is necessary to turn to 3D-3D correspondences to concurrently recover scale and pose parameters. In an effort to diminish its dependence on depth information, recent studies have introduced intriguing solutions that rely on multi-view NOCS maps [5] or the decoupled metric scale recovery strategy [6] to mitigate the scale ambiguity. We are very much interested in exploring these promising solutions to further improve the applicability of our method in future works. We sincerely appreciate your understanding and patience in this matter, and we are grateful for your insightful observations.
> > >
> > > Sincerely,
> > >
> > > Authors
> > >
> > > [1] Zhao et al., 3D-Aware Hypothesis & Verification for Generalizable Relative Object Pose Estimation, ICLR 2024.
> > >
> > > [2] Felice et al., Zero123-6D: Zero-shot Novel View Synthesis for RGB Category-level 6D Pose Estimation, arXiv 2024.
> > >
> > > [3] Li et al., NeRF-Pose: A First-Reconstruct-Then-Regress Approach for Weakly-supervised 6D Object Pose Estimation, ICCVW 2023.
> > >
> > > [4] Saxena et al., Generalizable Pose Estimation Using Implicit Scene Representations, ICRA 2023.
> > >
> > > [5] Chen et al., StereoPose: Category-Level 6D Transparent Object Pose Estimation from Stereo Images via Back-View NOCS, ICRA 2023.
> > >
> > > [6] Wei et al., RGB-based Category-level Object Pose Estimation via Decoupled Metric Scale Recovery, ICRA 2024.

---

> > > > ### Comment · Reviewer_heqf · 2024-08-12
> > > >
> > > > The authors have addressed my concerns, so I have raised my rating to 'Accept.' It is worth adding the discussion of those previous works to the main paper, accordingly.

---

> > > > > ### Author Response · Authors · 2024-08-13
> > > > > **Thank You!**
> > > > >
> > > > > Dear Reviewer heqf,
> > > > >
> > > > > Thank you for your feedback on our rebuttal. We are delighted that our response has addressed your questions. We will follow your suggestion to have a discussion on those previous works in our final version. We greatly appreciate your kind support for our work and are grateful for raising the score. Thank you very much!
> > > > >
> > > > > Sincerely,
> > > > >
> > > > > Authors

---

### Official Review · Reviewer_yRF6 · 2024-07-09

**Soundness:** 3
**Presentation:** 3
**Contribution:** 3
**Rating:** 7
**Confidence:** 4

**Summary:**

Authors introduce method for generalizable object pose estimation given RGB-D image. The approach builds upon DINOv2 model and adds two adapter blocks on top to accomplish better viewpoint estimation as well as object coordinate estimation. The resulting model is trained on synthetic data with contrastive loss combined with coordinate map loss. The model achieves good performance, while being generalizable to the unseen instances without any finetuning.

**Strengths:**

The paper presents a simple and working idea to extend DINOv2 model to category-level pose estimation by adding two adapter blocks on top. The method is then thoroughly evaluated and ablated, giving hints on what is important and what is not in the pipeline. The method also works well in the standard BOP setup on LINEMOD, which highlights the generalization capabilities of the method.

**Weaknesses:**

The Figure 1 is a bit misleading, while it is true that DINOv1 was not working out-of-the box for pose estimation, recent work such as [1] show that it can with some modifications. Why is DINOv1 used here and not much more performant DINOv2?

In the related work section authors omit some of the recent work on 6D pose estimation that are similar in nature to this method such as FoundPose [1]. This paper was released earlier than FoundationPose [2], however it is not discussed in this paper at all.

In comparison to BOP methods, the selection of baselines by authors is not fully thorough. For example, FoundationPose is mentioned however the method is not compared against it on LINEMOD.

At last, for the inference, no information about template preparation apart from L268-L269 on p. 6 is available. What kind of conditions are needed? What settings are used?

[1] Evin Pınar Örnek et al., FoundPose: Unseen Object Pose Estimation with Foundation Features, ArXiv:2311.18809

[2] Bowen Wen et al., FoundationPose: Unified 6D Pose Estimation and Tracking of Novel Objects, ArXiv:2312.08344

**Questions:**

1)	For pose estimation tasks the method for sampling rotations from SO(3) can very strongly affect resulting performance. I was wondering what kind of SO(3) sampling is used in generated synthetic data?

2)	Going to my last point in the weaknesses, can you please describe what exactly is the procedure to prepare the shape templates for the inference? What kind of setup is needed (textured vs. untextured, light sources, etc)?

**Limitations:**

Authors adequately discuss main limitation of this method, mainly related to occlusions and requirement of depthmaps for input images.

---

> ### Author Rebuttal · Authors · 2024-08-07
>
> >**Q1: Clarify Fig.1. Why is DINOv1 used in Fig.1 and not much more performant DINOv2?**
>
> **A1:** Thanks for pointing it out. In Fig.1, taking DINOv1 as an example, we aim to show that the pre-trained vision foundation model is not reliably used for category-level object pose estimation. This observation is indeed consistent across different representative vision foundation models, including CLIP, MVP, DINOv1, and DINOv2 (As shown in Fig.8 of the original paper). We attribute this to their limited 3D representation capability due to pre-training with text and 2D images. Based on this, we propose a foundation feature lifting module for more reliable object pose estimation, which has proven effective with different vision foundation models. To reduce the possible confusion, we will revise Fig.1 of the main paper to include results from different vision foundation models during revision.
>
> >**Q2: The recent work FoundPose[1] is not discussed in the paper.**
>
> **A2:** Thanks for your constructive suggestion. FoundPose[1] is a recent method for object pose estimation. Similar to FoundationPose[2], it aims to tackle the problem of instance-level unseen object pose estimation. The idea of integrating DINOv2 representation and bags-of-words descriptor is very impressive. During our submission, FoundPose is still an arXiv paper, while FoundationPose has been accepted by CVPR 2024. Therefore, we mainly focus on discussing published works while preparing the manuscript. We are also glad to find that FoundPose has been accepted by ECCV 2024 (the paper decision date is 1st July 2024). Note that our method is clearly distinguished from FoundPose by addressing a more challenging category-level estimation problem for unseen object categories. We will ensure to incorporate a discussion of FoundPose in the revised manuscript.
>
> >**Q3: Comparison with the recent BOP method FoundationPose[2].**
>
> **A3:** Thanks for your valuable suggestion. Considering that the main focus of our paper is category-level object pose estimation for unseen object categories, we compare our method with FoundationPose on the CO3D dataset. We utilized the official source code and pre-trained models from FoundationPose, adopting the model-based setup during inference. Both our method and FoundationPose use the same category-level shape templates for object pose estimation. The following table presents the evaluation results in terms of Acc.$15^\circ$ / Acc.$30^\circ$. Due to the character limit, we only report the per-category accuracy for 6 representative categories and the average accuracy across all 20 categories:
>
> |                | Motorcycle  | Backpack    | Bicycle      | Teddybear    | Car         | Chair           | Average         |
> | :--------------: | :-----------: | :-----------: | :------------: | :------------: | :------------: | :-------------: | :-------------: |
> | [3] | 42.4/55.1   | 13.3/23.1   | 25.3/46.9    | 20.4/43.8    | 39.4/54.2   | 59.3/67.7      | 30.1/48.8     |
> | Ours           | 56.4/76.3   | 30.6/47.4   | 46.5/59.2    | 25.2/54.4    | 55.6/74.4   |72.1/86.8  | **50.2/67.4**     |
>
> As can be observed, our method demonstrates its superiority over FoundationPose in challenging category-level object pose estimation for unseen object categories.
>
> >**Q4: The procedure of preparing the shape templates. What kind of conditions are needed and what settings are used to prepare the shape templates for the inference?**
>
> **A4:** Thanks for the comments. The procedure of preparing shape templates can be divided into the following three steps: (1) Normalize the input object CAD model to have a diameter of 1; (2) $N$ camera viewpoints ($N$ indicates the number of template images for inference) are sampled from a sphere centered on the normalized object CAD model; (3) Rendering the corresponding RGB-D template image at each viewpoint with Blender. Our method only needs an **untextured object CAD model**. During rendering shape templates, we **fix the position of the lighting source** on the top of the object, with a **random lighting color** that is uniformly sampled within $[0.5, 0.5, 0.5]$~$[1.0, 1.0, 1.0]$. We will append these implementation details in our revised manuscript.
>
> >**Q5: What kind of SO(3) sampling is used in generated synthetic data?**
>
> **A5:** Thanks for the comments. We customized the synthetic data generation pipeline[3] that has been widely used in the BOP challenge to generate our category-level synthetic data. To generate diverse synthetic data, two aspects of pose sampling are involved: (1) **Object on-surface sampling.** The object would be placed upright onto a plane of the synthetic scene, and the in-plane position and orientation of the object would be randomly sampled. (2) **Camera pose sampling.** The camera location is first sampled around the object using the "uniform_elevation" sampling used in BOP. Then, the camera rotation is further determined by a point of interest which is randomly sampled from the scene, plus a sampled camera in-plane rotation within a specified range ([$-30^\circ$, $30^\circ$] used in our synthetic data). We will append these details related to synthetic data sampling in our revised manuscript.
>
> [1] Örnek et al. FoundPose: Unseen Object Pose Estimation with Foundation Features. ECCV 2024.
>
> [2] Wen et al. FoundationPose: Unified 6D Pose Estimation and Tracking of Novel Objects. CVPR 2024.
>
> [3] Denninger et al. BlenderProc: Reducing the Reality Gap with Photorealistic Rendering. RSS Workshop 2020.

---

> > ### Comment · Reviewer_yRF6 · 2024-08-11
> >
> > I thank authors for addressing questions and concerns of all reviewers. I believe with the questions addressed, this is a good paper and I'm updating my score to "Accept".

---

> > > ### Author Response · Authors · 2024-08-12
> > > **Thank You!**
> > >
> > > Dear Reviewer yRF6,
> > >
> > > Thank you for your feedback on our rebuttal. We are pleased that our response has addressed your questions. Much appreciate your kind support for our work. Thanks a lot!
> > >
> > > Sincerely,
> > >
> > > Authors

---

### Official Review · Reviewer_P7Hz · 2024-07-23

**Soundness:** 3
**Presentation:** 3
**Contribution:** 3
**Rating:** 6
**Confidence:** 2

**Summary:**

This paper addresses the task of category-level object pose estimation for unseen object categories from paired RGB-D imagery. To deal with unseen object categories, the authors leverage a vision-foundation model (Dino-v2 in this case).
The entire pipeline works in two stages. Given an RGB-D input, the first step retrieves reference images with the closest cosine similarity with respect to the object features. This is done in order to retrieve the viewpoint R, t of the reference images. To enhance the 3D representation of this step, the authors introduce a 2D-to-3D feature lifting step. In a nutshell, this step utilizes 3 positional information from the object's point cloud and merging these with the foundation features and improve the retrieval step.
The second step performs the actual object pose estimation. The process involves transforming the query and reference object shapes into a normalized space and encoding them using a point cloud Transformer integrated with pre-trained image features. The final NOCS coordinates for the query are calculated using a softmax function on the product of the shape embeddings of both objects, combined with the reference's NOCS coordinates. Training the model was enhanced with Blender-generated synthetic data.
Experimental results demonstrate large improvements over SotA on category-level unseen object pose estimation (Tbl. 1, 2). Ablative studies motivate the use of the proposed modules.

**Strengths:**

- The approach makes clever use of a foundation model and improves its shortcomings via 3D-aware feature transformation
- The approach greatly outperforms state-of-the-art on category-level unseen object pose estimation. It can also be used for instance-level unseen object pose estimation and remains competitive.
- I greatly appreciated the evaluation in open-world scenarios (Sec. 4.5) which demonstrates its use in real-world scenario and not just on isolated benchmarks.
- Ablative studies demonstrating the performance using various foundation models is informative in showing how the performance of a foundation model correlates with the downstream tasks.

**Weaknesses:**

- The method relies on RGB images paired with depth images. This requires specialized sensors and reduces the applicability in the real world. I would've appreciated an ablative study in which RGB images alone was input into a depth-estimation network before passing it into VFM-6D. This could greatly enhance the strengths of the paper and potentially show that RGB-only predictions would be possible as well.
- The authors mentioned that the model is limited in the presence of severe occlusion. A plot show-casing this (e.g percentage of occlusion vs accuracy) would be beneficial to demonstrate the robustness/weakness of the model with respect to this aspect.

**Questions:**

- VFM-6D largely dominates in terms of performance compared to related work. Yet in Tbl. 1 it performs worse in two ranges. Is this solely due to the seen/unseen difference? Since VFM-6D outperforms on the larger thresholds, what would the authors propose would push VFM-6D to be SotA on all thresholds?

**Limitations:**

The authors were very clear on the limitations of the model

---

> ### Author Rebuttal · Authors · 2024-08-07
>
> >**Q1: Ablation study on RGB-only object pose estimation.**
>
> **A1:** Thanks for your suggestion. Due to the character limit, please refer to **Q1** in the global response for detailed evaluation results. As can be observed, the RGB-only variant achieves comparable pose accuracy on average when compared with the original RGBD-based performance. These results indicate the application potential of our method in scenarios that lack depth observation. Please also refer to Fig.1 of the PDF file uploaded in the global response for detailed qualitative visualization results.
>
> >**Q2: Results under different percentages of occlusion.**
>
> **A2:** Thanks for the good suggestion. First, we evaluated our method under different levels of occlusion on the LineMOD-Occlusion dataset. The table below reports the average recall of ADD(-s).
> |         | No occlusion | <30% | 30%~60% | >60% |
> | :-------: | :------------: | :--------------: | :-----------------: | :--------------: |
> | ADD-(s) | 90.3         | 87.4           | 69.4              | 32.4           |
>
>
> Second, we conducted more evaluations on the CO3D dataset. Since scenes from CO3D are generally occlusion-free, we randomly masked out different percentages of image regions to mimic the effect of object occlusion. The table below reports the Acc.$15^{\circ}$ / Acc.$30^{\circ}$ results on 5 representative categories of CO3D.
>
> |            | No occlusion |    <30%   |  30%~60%  |   >60%    |
> | :----------: | :------------: | :---------: | :---------: | :---------: |
> | Motorcycle | 56.4/76.3    | 54.9/70.1 | 43.3/64.1 | 32.3/51.0 |
> | Chair      | 72.1/86.8    | 68.2/82.0 | 55.3/74.1 | 38.4/57.7 |
> | Bicycle    | 46.5/59.2    | 45.9/60.5 | 42.9/61.6 | 39.0/60.2 |
> | ToyPlane   | 55.1/66.6    | 53.0/64.3 | 44.8/56.4 | 45.0/55.5 |
> | ToyTrain   | 61.9/80.2    | 61.6/77.1 | 46.3/76.6 | 40.9/71.1 |
>
>
> These results show that our method is relatively robust to small and moderate occlusion percentages, and would suffer a performance drop when facing severe occlusions larger than 60% occlusion rate. Moreover, as we have discussed in the main paper, in practical application scenarios, this occlusion issue could be potentially addressed by active perception and mobile manipulation to find an occlusion-free viewpoint.
>
> >**Q3: Clarify results presented in Table 1.**
>
> **A3:** Sorry for any confusion caused. The evaluation of Wild6D contains comparisons with two groups of approaches. To improve the clarity, we have reorganized the original Table 1 into these two distinct parts to better communicate the Wild6D evaluation findings:
>
> - **Comparison with conventional category-level approaches.** For the competing methods SPD, SGPA, DualPoseNet and GPV-Pose, we first trained their models on the training data containing the testing categories, and then tested their performance on Wild6D. For our method, we did not train it on data containing the testing categories, and directly tested its performance on Wild6D. The following table presents the comparative results. Conventional category-level approaches highly rely on category-specific training. They have to train and test their models on the same sets of categories. In this case, Wild6D is **not unseen** to them. In contrast, without training on the five testing categories, our method significantly outperforms these category-level approaches in $10^\circ2cm$ and $10^\circ5cm$ accuracy, and achieves comparable accuracy under more strict thresholds.
>
>   |             | $5^\circ2cm$ | $5^\circ5cm$ | $10^\circ2cm$ | $10^\circ5cm$ |
>   | :---------: | :----------: | :----------: | :-----------: | :-----------: |
>   |     SPD     |     2.6      |     3.5      |      9.7      |     13.9      |
>   |    SGPA     |   **20.1**   |   **27.8**   |     29.0      |     39.4      |
>   | DualPoseNet |     17.8     |     22.8     |     26.3      |     36.5      |
>   |  GPV-Pose   |     14.1     |     21.5     |     23.8      |     41.1      |
>   |    Ours     |     19.3     |     21.6     |   **34.9**    |   **44.2**    |
>
> - **Comparison with category-agnostic approaches.** The following table presents the comparative results. Note that all competing methods are not trained on testing categories. In this case, categories in Wild6D are **unseen** to all three methods. Our method outperforms the other two competing methods in all metrics.
>
>   |              | $5^\circ2cm$ | $5^\circ5cm$ | $10^\circ2cm$ | $10^\circ5cm$ |
>   | :----------: | :----------: | :----------: | :-----------: | :-----------: |
>   | PoseContrast |     2.3      |     4.7      |      5.5      |     10.1      |
>   |     ZSP      |     9.6      |     12.1     |     16.6      |     23.0      |
>   |     Ours     |   **19.3**   |   **21.6**   |   **34.9**    |   **44.2**    |

---

> > ### Comment · Reviewer_P7Hz · 2024-08-12
> > **response to rebuttal**
> >
> > I thank the authors for providing their rebuttal and clarifying all the questions I had. I suggest the authors include these additional results into the main paper as these are informative.
> >
> > I think this is a solid paper and vote for acceptance as before.

---

> > > ### Author Response · Authors · 2024-08-13
> > > **Thank You!**
> > >
> > > Dear Reviewer P7Hz,
> > >
> > > Thanks for your feedback on our rebuttal. We are glad that our response has addressed your questions. We will follow your suggestion to include those informative experimental results in our final version. We truly appreciate your kind support for our work. Thanks a lot!
> > >
> > > Sincerely,
> > >
> > > Authors

---

### Author Rebuttal · Authors · 2024-08-07

We are grateful to all reviewers for taking the time to review and provide constructive feedback. We are glad to see that most reviewers recognized the novelty of our method, the strength of our experimental evaluations, and the good presentation of our paper:
- “Presents a technically sound method to tackle a challenging but important problem.” (heqf)
- “Makes clever use of a foundation model and greatly outperforms state-of-the-art methods.” (P7Hz)
- “The method is thoroughly evaluated and ablated, highlighting its good generalization capability.” (yRF6)
- “Evaluation in open-world scenarios is greatly appreciated, demonstrating its application in real-world scenarios and not just on isolated benchmarks.” (P7Hz and heqf)

**(Q1: RGB-based ablation study.)** We also noticed that some reviewers provided good suggestions to evaluate our method under RGB-only setting (P7Hz and heqf). We followed the suggestion to conduct this ablation study. Specifically, we input the RGB image into the pre-trained depth-anything[1] model to recover the corresponding metric depth. Then, we input the RGB image and the recovered depth into our VFM-6D for object pose estimation. Without additional training, the table below reports Acc.$15^{\circ}$ / Acc.$30^{\circ}$ results on CO3D dataset.

|       | Motorcycle  | Backpack    | Bicycle      | Teddybear    | Book         | Car           | Chair       |
| ----- | ----------- | ----------- | ------------ | ------------ | ------------ | ------------- | ----------- |
| RGB-D | 56.4/76.3   | 30.6/47.4   | 46.5/59.2    | 25.2/54.4    | 41.9/43.5    | 55.6/74.4     | 72.1/86.8   |
| RGB-only   | 49.7/68.4   | 26.4/48.5   | 44.6/52.5    | 29.0/56.9    | 38.5/39.7    | 43.1/63.3     | 81.5/94.4   |
|       | **Handbag** | **Hydrant** | **Keyboard** | **Mouse**    | **Toaster**  | **Hairdryer** | **Laptop**  |
| RGB-D | 75.5/89.5   | 35.4/91.6   | 57.1/57.1    | 38.3/57.3    | 44.2/47.7    | 63.0/85.2     | 96.8/97.5   |
| RGB-only   | 40.2/77.3   | 61.2/98.9   | 29.4/48.8    | 23.9/52.5    | 39.1/43.9    | 52.8/70.6     | 85.2/98.2   |
|       | **Remote**  | **Toilet**  | **ToyBus**   | **ToyPlane** | **ToyTrain** | **ToyTruck**  | **Average** |
| RGB-D | 33.3/36.8   | 47.1/75.1   | 24.8/53.4    | 55.1/66.6    | 61.9/80.2    | 43.4/67.8     | 50.2/67.4   |
| RGB-only   | 22.5/39.2   | 53.5/80.1   | 21.3/46.9    | 46.0/63.2    | 51.9/79.5    | 37.7/62.0     | 43.9/64.2   |

As can be observed, our method can effectively leverage the depth map estimated from the RGB image for generalizable object pose estimation. Note that in the context of category-level object pose estimation, we usually jointly estimate object size and pose parameters. Based on the estimated metric depth map, our method can recover object rotation precisely and can recover object size and translation up to a global scale factor. The RGB-only variant achieves comparable rotation accuracy on average when compared with the original RGBD-based performance. These promising results indicate the application potential of our method in scenarios that lack depth observation. Please also refer to Fig.1 of the uploaded PDF file for detailed qualitative pose prediction results.

[1] Yang et al. Depth Anything: Unleashing the Power of Large-Scale Unlabeled Data. CVPR 2024.

Below we further reply to each reviewer's comments point-by-point. We hope that our rebuttal can successfully address the reviewers' questions, and we look forward to receiving the reviewers’ support for our work.

---

### Decision · Program_Chairs · 2024-09-25

**Decision:**

Accept (poster)

**Comment:**

This paper tackles the challenging yet important task of category-level pose estimation for unknown object categories. The proposed method effectively addresses this issue by leveraging vision-and-language foundation models to achieve good object representation and robust shape matching. The training is conducted using cost-effective synthetic data and has been evaluated on various benchmark datasets and real-world scenarios. All reviewers have highly valued the reliability of the experimental results, which include comparisons with state-of-the-art methods and comprehensive ablation studies, as well as the impact of experiments in open-world scenarios. Regarding the idea of utilizing foundation models, one reviewer praised the approach, while another questioned the novelty in comparison to other similar approaches. However, this concern was thoroughly addressed in the authors’ rebuttal. Although many technical questions were raised, ultimately, all reviewers were satisfied with the authors' responses, and three out of four reviewers raised their scores. It is encouraged that the detailed analysis and additional experimental results provided in the authors’ rebuttal will be reflected in the final manuscript.